# Isotope discrimination of carbonyl sulfide (<sup>34</sup>S) and carbon dioxide (<sup>13</sup>C, <sup>18</sup>O) during plant uptake in flow-through chamber experiments

Sophie L. Baartman<sup>1,2</sup>, Steven M. Driever<sup>3</sup>, Maarten Wassenaar<sup>4</sup>, Linda M.J. Kooijmans<sup>2</sup>, Nerea Ubierna<sup>6</sup>, Leon Mossink<sup>3</sup>, Maria E. Popa<sup>2</sup>, Ara Cho<sup>2</sup>, Lisa Wingate<sup>6</sup>, Thomas Röckmann<sup>1</sup>, Steven M.A.C. van Heuven<sup>5</sup>, Maarten C. Krol<sup>1,2</sup>

<sup>1</sup>Institute for Marine and Atmospheric Research Utrecht (IMAU), Princetonplein 5, 3584 CC Utrecht, The Netherlands

<sup>2</sup>Meteorology and Air Quality, Wageningen University and Research Centre, Droevendaalsesteeg 4, 6708 PB Wageningen, The Netherlands

<sup>3</sup>Centre for Crop Systems Analysis, Wageningen University and Research Centre, Droevendaalsesteeg 4, 6708 PB Wageningen, The Netherlands

<sup>4</sup>Horticulture and Product Physiology, Wageningen University and Research Centre, Droevendaalsesteeg 4, 6708 PB Wageningen, The Netherlands

5 Centre for Isotope Research (CIO), Energy and Sustainability Research Institute Groningen, University of Groningen, 9747 AG Groningen, the Netherlands

<sup>6</sup>Institut National de la Recherche Agronomique, Bordeaux Sciences Agro, Unité Mixte de Recherche 1391, Interactions Sol-Plante-Atmosphère, 33140 Villenave d'Ornon, France

Correspondence to: Sophie L. Baartman (sophie.baartman@wur.nl)

5

10

**Abstract.** Carbonyl sulfide (COS) has been proposed as a proxy for gross primary production (GPP), as it is taken up by plants through a pathway comparable to that of CO<sub>2</sub>. COS diffuses into the leaf, where it undergoes an essentially one-way reaction in the mesophyll cells, catalyzed by the enzyme carbonic anhydrase (CA), and is likely not respired by the leaf. In order to use COS as a proxy for GPP, the mechanisms of COS uptake and its coupling to photosynthesis need to be well understood. Characterizing the isotopic discrimination of COS during plant uptake could provide valuable information on the physiological COS uptake process and may help to constrain the COS budget.

This study presents joint measurements of isotope discrimination during plant uptake for COS (CO<sup>34</sup>S) and CO<sub>2</sub> (<sup>13</sup>CO<sub>2</sub> and C<sup>18</sup>O<sup>16</sup>O). A C<sub>3</sub> plant, sunflower (*Helianthus annuus*), and a C<sub>4</sub> plant, papyrus (*Cyperus papyrus*), were enclosed in a flow-through plant chamber and exposed to varying light levels. The incoming and outgoing gas compositions were measured online, and discrete air samples were taken for isotope analysis. Simultaneously measuring fluxes and isotope discrimination of both COS and CO<sub>2</sub> yielded a unique dataset that includes information on the plant's behavior and allowed for the estimation of stomatal- and mesophyll conductance.

The average COS uptake fluxes were  $73.3 \pm 1.5$  pmol m<sup>-2</sup> s<sup>-1</sup> for sunflower and  $107.3 \pm 1.5$  pmol m<sup>-2</sup> s<sup>-1</sup> for papyrus (PAR > 0) and displayed virtually no trend with increasing PAR from 200 to 600  $\mu$ mol m<sup>-2</sup> s<sup>-1</sup>. The mean observed  $^{34}\Delta$  for COS was  $3.4 \pm 1.0$  % for sunflower and  $2.6 \pm 1.0$  % for papyrus.  $^{34}\Delta$  was stable across all light

intensities, which could be explained by a sufficient stomatal opening and low variability in the ratio of mesophyll vs. ambient COS mole fraction,  $C_m^S/C_a^S$ . For CO<sub>2</sub>, a negative relationship was observed between the uptake flux and the isotopic discriminations  $^{13}\Delta$  and  $^{18}\Delta$ . The CO<sub>2</sub> uptake and  $^{13}\Delta$  and  $^{18}\Delta$  values indicate that the sunflower behaved as expected for a C<sub>3</sub> plant, while the low CO<sub>2</sub> flux and high  $^{13}\Delta$  and  $^{18}\Delta$  values observed for papyrus were not in the typical C<sub>4</sub> range, which was perhaps due to the relatively low light conditions during our experiments.

### 1. Introduction

Photosynthetic uptake of carbon dioxide (CO<sub>2</sub>) by the terrestrial biosphere, quantified by the gross primary production (GPP), is the largest sink of atmospheric CO<sub>2</sub>, and may be altered as the climate changes (Friedlingstein et al., 2023). For making accurate future climate projections, it is important to quantify changes in the functioning of the biosphere and its influence on the atmospheric composition. Several techniques can be used to quantify photosynthesis and respiration fluxes at the ecosystem- and larger scales, such as Eddy Covariance (EC) (Asaf et al., 2013; Billesbach et al., 2014; Commane et al., 2015; Wehr et al., 2017; Vesala et al., 2022) or variations in the stable isotopic composition of CO<sub>2</sub> (e.g. Farquhar and Lloyd, 1993; Farquhar et al., 1993; Wingate et al., 2007; Gentsch et al., 2014; Wehr et al., 2015;). However, these techniques have limitations, because they either measure net CO<sub>2</sub> fluxes (Wohlfahrt et al., 2012; Kooijmans et al., 2017) or they require additional measurements such as the oxygen isotopic composition of water pools (Wingate et al., 2010; Adnew et al., 2020). Because of these limitations, other potential independent proxies for GPP have recently gained attention, especially the trace gas carbonyl sulfide (COS or OCS, COS henceforth) (Sandoval-Soto et al., 2005; Montzka et al., 2007; Campbell et al., 2008; Whelan et al., 2018; Lai et al., 2024).

COS is the most abundant sulfur-containing atmospheric trace gas, with a tropospheric mole fraction of around 500 pmol mol<sup>-1</sup> that displays a strong seasonal cycle, mostly due to the uptake of COS by terrestrial vegetation during photosynthesis. Figure 1 shows a schematic of the uptake pathways and assimilation locations of COS and CO<sub>2</sub> in the leaf. Similarly to CO<sub>2</sub>, COS diffuses across the leaf boundary layer, through the stomata and into the leaf mesophyll cells (Protoschill-Krebs and Kesselmeier, 1992; Protoschill-Krebs et al., 1996). There, COS is hydrolyzed in an essentially one-way reaction, catalyzed by the enzyme carbonic anhydrase (CA), in contrast to the reversible hydration reaction that CO<sub>2</sub> undergoes (Protoschill-Krebs and Kesselmeier, 1992; Protoschill-Krebs et al., 1996). Assuming that there is no COS emission, the COS uptake by plants is proportional to photosynthetic uptake of CO<sub>2</sub>, and therefore, GPP can be derived from the leaf-scale relative uptake ratio (LRU) of COS and CO<sub>2</sub> uptake fluxes,  $A^S$  (pmol m<sup>-2</sup> s<sup>-1</sup>) and  $A^C$  (µmol m<sup>-2</sup> s<sup>-1</sup>), normalized to their atmospheric mole fractions,  $C_a^S$  (pmol mol<sup>-1</sup>) and  $C_a^C$  (µmol mol<sup>-1</sup>) using Eq. (1):

$$LRU = \frac{A^S}{A^C} * \frac{C_a^C}{C_a^S} \tag{1}$$

If we assume negligible daytime leaf respiration, or if we account for it,  $A^{C}$  can be replaced by GPP, which can then be estimated using Eq. (2) (re-arrangement of Eq. (1)) (Campbell et al., 2008).

$$GPP = A^{S} \frac{C_a^{C}}{C_s^{S}} * \frac{1}{LRU}$$
 (2)

While the use of LRU as a link between COS and CO<sub>2</sub> fluxes seems promising, some studies have shown that the LRU is not constant among species and changes with environmental conditions such as photosynthetically active radiation (PAR), temperature and vapor pressure deficit (VPD) (Kooijmans et al., 2019; Maignan et al., 2021; Sun et al., 2022; Spielmann et al., 2023; Sun et al., 2024). Additionally, the existence of a COS compensation point suggests that emissions can occur for some species under certain circumstances (Goldan et al., 1988; Kesselmeier and Merk, 1993; Kuhn and Kesselmeier, 2000; Maseyk et al., 2014; Belviso et al., 2022) Thus, a more thorough understanding of the physiological drivers and limitations of COS uptake by plants, and its relationship with CO<sub>2</sub> uptake, is needed.

Figure 1. Schematic (simplified) representation of the diffusion pathways (zigzag lines) of  $CO_2$  (left) and COS (right) into a  $C_3$  leaf, with the conductance parameters being boundary layer-  $(g_{bl})$ , stomatal-  $(g_s)$  and mesophyll conductance  $(g_m)$ . The  $CO_2$  and COS mole fractions are indicated as  $C_a$  (atmospheric),  $C_i$  (intercellular space),  $C_m$  (mesophyll cell) and, for  $CO_2$ ,  $C_c$  indicates the mole fraction in the chloroplast (the green, bordered area). The enzymes ribulose-1,5-biphosphate carboxylase oxygenase (RuBisCo, inside the chloroplast) and carbonic anhydrase (CA, right figure only) catalyze  $CO_2$  and COS fixation. The purple line represents the mesophyll cell wall, and the blue line indicates the plasma membrane.

Using the distinct fingerprints of chemical and diffusion processes, the isotopic fractionation of COS during plant uptake could be used to help improve understanding of processes driving COS plant uptake. For example, isotope measurements may provide insights on the role of environmental factors, such as PAR and VPD with respect to LRU variations. Improved global estimates of isotope discrimination of C<sub>3</sub> and C<sub>4</sub> species may then be used to better constrain the COS budget (Davidson et al., 2022) and possibly aid in improving the COS-derived GPP estimate.

Isotope studies on COS uptake build on the extensive experience and literature on the isotope effects associated with the uptake of CO<sub>2</sub>. The discrimination against CO<sup>34</sup>S (‰) is defined in Eq. (3), where <sup>32</sup>k and <sup>34</sup>k are the reaction rate coefficients for uptake of CO<sup>32</sup>S and CO<sup>34</sup>S, respectively:

$$^{34}\Delta = 1 - \frac{^{34}k}{^{32}k}. (3)$$

Isotope discrimination occurs both during diffusion of COS into the leaf and due to the preferential hydrolysis of lighter isotopologues by CA (Davidson et al., 2022). Similar to the model developed by Farquhar et al. (1982) for <sup>13</sup>CO<sub>2</sub> discrimination during photosynthesis and as the reaction with CA is supposed to be irreversible (Protoschill-

Krebs and Kesselmeier, 1992; Protoschill-Krebs et al., 1996), the net  $CO^{34}S$  discrimination during plant uptake ( $^{34}\Delta$ ) can be expressed as a function of the ratio of COS mole fraction at the site of assimilation (the end-point), in the mesophyll cell ( $C_m^S$ ) versus the COS mole fraction in ambient air ( $C_a^S$ ) (Davidson et al., 2022):

$$^{34}\Delta = \bar{a} + (h - \bar{a})\frac{c_m^S}{c_n^S},\tag{4}$$

where  $\bar{a}$  is the fractionation occurring during diffusion of COS into the leaf up to the mesophyll cell, which incorporates leaf boundary layer (BL) diffusion, stomatal diffusion and gas-liquid interface dissolution and diffusion, and h is the S isotope fractionation during fixation by the enzyme carbonic anhydrase (CA).

 $C_m^S$  has been suggested to be close to zero in C<sub>3</sub> plants (Stimler et al., 2011; Stimler et al., 2012). When  $C_m^S$  = 0, Eq. (4) reduces to  ${}^{34}\Delta = \bar{a}$ , thus  ${}^{34}\Delta$  is caused solely by diffusion differences between CO<sup>32</sup>S and CO<sup>34</sup>S ( $\bar{a}$ ) through the stomata and up to the mesophyll. Binary molecular diffusion of COS in air is theoretically expected to provide a  ${}^{34}\Delta$  value of around 5 ‰, because of the differences in molecular masses between the different COS isotopologues (Angert et al. 2019). However, this may be a too crude simplification of the diffusion processes taking place, as COS diffusion not only involves gaseous diffusion but also gas-liquid interface diffusion from the intercellular space to the mesophyll cell (Fig. 1) (Stimler et al., 2010; Berry et al., 2013). When including stomatal diffusion, leaf BL diffusion, and gas-liquid phase diffusion in the mesophyll cell, Davidson et al. (2022) calculated an overall diffusion fractionation value of  $\bar{a} = 1.6 \pm 0.1\%$  for  ${}^{34}S$ .

Still, it is not known whether the COS mole fraction in the mesophyll always reaches values close to zero, especially for C<sub>4</sub> species, in which CA activity is low (Stimler et al., 2011). In the case of non-zero  $C_m^S$ , values for the enzymatic fractionation during COS fixation by CA (h) are needed to calculate <sup>34</sup> $\Delta$ . Davidson et al. (2022) determined an enzymatic fractionation for <sup>34</sup>S, h, of 15 ± 2 % from experiments in which the plants were exposed to high CO<sub>2</sub> (2900 ± 90 pmol mol<sup>-1</sup>) and COS (3.4 ± 0.1 µmol mol<sup>-1</sup>) mole fractions.

In another set of experiments by Davidson et al. (2022), this time using ambient  $CO_2$  (500  $\pm$  80 pmol mol<sup>-1</sup>) and COS (0.53  $\pm$  0.02 nmol mol<sup>-1</sup>) mole fractions, their observed <sup>34</sup> $\Delta$  values were 1.6  $\pm$  0.1 ‰ for  $C_3$  and 5.4  $\pm$  0.5 ‰ for  $C_4$  species. These authors attributed the higher discrimination value for  $C_4$  species to the lower CA activity, which could lead to a non-zero COS mole fraction at the site of CA and discrimination by this enzyme.

As the methodology for isotope ratio measurements of COS has only recently been established (Hattori et al., 2015; Angert et al., 2019; Baartman et al., 2022), the only studies that have determine COS isotope discrimination during plant uptake are by Davidson et al., (2021) and Davidson et al., (2022). These studies used a closed-chamber approach and, as mole fractions of CO<sub>2</sub>, COS and H<sub>2</sub>O change during experiments with closed chambers, there is a potential risk that feedback processes on stomatal conductance and other metabolic processes may have contributed to the observed discrimination. Hence, these results may not reflect typical leaf conditions. With flow-through chambers, conditions can be monitored online and kept stable throughout the entire experiment, also allowing for easier repetition of the experiments.

In this work, we introduce a new methodoly for measuring COS isotope discrimination in plants, using a flow-through plant chamber, which was closely monitored to maintain stable conditions. We demonstrate the advantages of simultaneously measuring COS and CO<sub>2</sub> fluxes, and isotope discrimination of COS uptake against  $CO^{34}$ S and  $CO_{2}$  uptake against  $CO^{34}$ S and  $CO^{34}$ S and

discrimination against <sup>13</sup>CO<sub>2</sub> (<sup>13</sup>Δ) can be used to explain variations in photosynthesis rates and to estimate stomatal conductance (Farquhar & Richards, 1984; Farquhar et al., 1989; Cernusak et al., 2013). During photosynthesis, CO<sub>2</sub> can exchange oxygen atoms with the leaf water, catalyzed by CA, and partly diffuse back to the atmosphere with changed isotopic composition. The resulting apparent discrimination against <sup>12</sup>C<sup>16</sup>O<sup>18</sup>O (<sup>18</sup>Δ) during photosynthesis can serve as a proxy for gross biosphere-atmosphere CO<sub>2</sub> exchange (Francey and Tans; Yakir, 1998; Adnew et al., 2020). Both <sup>13</sup>Δ and <sup>18</sup>Δ display a typical and distinct range of values for C<sub>3</sub> and C<sub>4</sub> species and depend on environmental factors (Farquhar et al., 1982; Stimler et al., 2011; Adnew et al., 2020). Therefore, the joint COS and CO<sub>2</sub> measurements allowed investigating the relationship between COS and CO<sub>2</sub> isotope effects, where the CO<sub>2</sub> data provide additional information for validating the experimental setup and the plant behavior.

#### 2. Methods

#### 2.1. Plant materials and growing conditions

Experiments were conducted with one C<sub>3</sub> plant, sunflower (*Helianthus annuus* "Sunsation"), and an assemblage of stems and leaves from the C<sub>4</sub> plant papyrus (*Cyperus papyrus*). A sunflower in the flowering stage was obtained at a local garden center. A large papyrus shrub was available and grown at the tropical greenhouse at Wageningen Univesity and Research (WUR). Three large stems with leaves were carefully cut from this larger shrub, using a sharp razor, and transported in water to the lab, where they were kept in water throughout the chamber measurements. The sunflower plant and papyrus cuttings were kept under a lamp with a solar-like spectrum (*ca.* 400 μmol m<sup>-2</sup> s<sup>-1</sup> PAR, LED growth light SMD2835, Ortho, China) before experiments started and watered sufficiently before and during the measurements. Leaf surface area of sunflower and papyrus were measured after the experiments using a LI-3100 (Li-Cor, Lincoln, NE, USA). This instrument was calibrated using a metal disk with a surface area of exactly 50.00 cm<sup>2</sup>.

#### 2.2. Whole plant gas exchange system

Gas exchange experiments were conducted at Wageningen University and Research (WUR) using a custom-built whole plant chamber that was developed for estimating net photosynthetic CO<sub>2</sub> assimilation and transpiration (Lazzarin et al., 2024). The main component is a flow-through plant chamber, which can be fed with different gas mixtures. Two analyzers were used to measure in- and outgoing mole fractions and we used an add-on module for discrete air samples (Fig 2.).

Figure 2. Schematic overview of the setup to determine CO<sub>2</sub> and COS photosynthetic isotope discrimination by coupling a custom-built plant chamber to a LI-7000, a QCLS and a system to fill up gas canisters for posterior isotope analysis with IRMS. MFC: mass flow controller; QCLS: Quantum Cascade Laser Spectrometer. CO<sub>2</sub> and COS were mixed into humidified synthetic air and introduced into the plant chamber. The in- and outflowing airstreams of the chamber (air<sub>in</sub> and air<sub>out</sub>) were measured by both the LI-7000 and QCLS instruments. Air was dried using Mg(ClO<sub>4</sub>)<sub>2</sub> before the QCLS and when taking a sample for isotope analysis.

The plant chamber was made of clear plexiglass lined with a FEP foil (Holscot Europe, Breda NL) to prevent water from sticking to the chamber walls. The chamber had a diameter of 29 cm, and the height was either 18 or 27 cm, depending on the plant size. To ensure proper air mixing and leaf boundary layer reduction, three SanAce40W ventilators (type 9WL0424P3J001, Sanyo120 Denki, Philippines) were placed in a circular pattern at the bottom of the chamber. Fan speed was controlled with a SanAce PWM controller. The entire chamber was placed inside a 63x63 cm² enclosure with white reflective walls that ensured uniform horizontal light distribution. Air temperature inside the plant chamber was measured with a LM35 temperature sensor (Texas Instruments). Temperature of the plant chamber was controlled using heating cables positioned around the outside of the plant chamber (in combination with a PID controller) and two 12V computer fans were used to provide airflow and cooling around the plant chamber. Light was

provided by LED lighting mounted above the chamber with a spectrum resembling sunlight (artificial sunlight research modules generation 2, Specialty Lighting Holland B. V., Breda, the Netherlands). PAR was quantified during the experiments just above the chamber using a handheld PAR sensor (LI-190, Li-Cor, Lincoln, NE, USA). Plants were placed in the chamber, and the bottom two plexiglass panels were closed around the stem of the plant and sealed it with Terostat RB VII, ensuring that the plant was isolated from the soil or water (in the case of the papyrus), and making sure the chamber was leak-free. Two pictures of the plant chamber are shown in Appendix A, Fig. A2.

Synthetic air humidified with a temperature-controlled water bubbler (dew point temperature 17 °C) was mixed with pure CO<sub>2</sub> using mass flow controllers (MFC), to reach the desired CO<sub>2</sub> and H<sub>2</sub>O mole fractions. Subsequently, COS from a cylinder with 700 nmol mol<sup>-1</sup> COS in synthetic "zero" air was supplied to the mix using a MFC to establish the target COS mole fractions of approximately 2 nmol mol<sup>-1</sup>. The flow rate of the total (combined) air mixture into the chamber was controlled by a MFC to around 8 L min<sup>-1</sup>, depending on the experiment conducted. The COS and CO<sub>2</sub> isotopic composition of the ingoing air was determined using the methods described in 2.5 and the values are provided in Table 1.

Table 1. Isotope composition of the inlet gas (air<sub>in</sub>) supplying the plant chamber determined from samples collected in canisters and analyzed with IRMS. Values are reported on the Vienna Canyon Diablo Troilite (VCDT) ( $\delta^{34}$ S), the Vienna Pee Dee Belemnite (VPDB) ( $\delta^{13}$ C) and Vienna Standard Mean Ocean Water (VSMOW) ( $\delta^{18}$ O) scales.

| Plant     | δ <sup>34</sup> S COS VCDT (‰) | δ <sup>13</sup> C CO <sub>2</sub> VPDB (‰) | δ <sup>18</sup> O CO <sub>2</sub> VSMOW (%) |
|-----------|--------------------------------|--------------------------------------------|---------------------------------------------|
| Sunflower | $11.9 \pm 1.2$                 | $-23.1 \pm 0.1$                            | $15.5 \pm 0.1$                              |
| Papyrus   | $12.1 \pm 0.5$                 | $-23.0 \pm 0.1$                            | $15.9 \pm 0.1$                              |

The CO<sub>2</sub> and H<sub>2</sub>O mole fractions of both the in-going air (air<sub>in</sub>, reference line) and the outgoing air (air<sub>out</sub>, sample line) of the chamber were analyzed with a LI-7000 infrared gas analyzer (LI-COR Biosciences, Lincoln, Nebraska, USA). To measure the COS mole fractions of air<sub>in</sub> and air<sub>out</sub>, we used a quantum cascade laser spectrometer (QCLS, TILDAS, Aerodyne Inc, USA) from the Center for Isotope Research, Rijksuniversiteit Groningen (CIO-RUG). This instrument also measured CO<sub>2</sub> mole fractions, which were validated with the readings of the LI-7000 and used for further analyses. QCLS used a 50 mL min<sup>-1</sup> flow and was manually switched between air<sub>in</sub>, air<sub>out</sub> and calibration cylinders. The air entering the QCLS was dried with magnesium perchlorate (Mg(ClO<sub>4</sub>)<sub>2</sub>) dryers. Calibration of the QCLS was performed at least twice a day using the working standards from the CIO-RUG, which are calibrated against NOAA-certified cylinders. Possible instrumental baseline drift during the experiments was corrected by measuring pure nitrogen (N<sub>2</sub>) multiple times during the experiment. For a detailed description of the QCLS instrument and calibration procedures, see Kooijmans et al. (2017). Blank measurements with an empty chamber were performed before a plant was installed in the chamber to ensure that the COS, CO<sub>2</sub> and H<sub>2</sub>O mole fractions of air<sub>in</sub> and air<sub>out</sub> were equal.

Samples for isotope analysis of COS and CO<sub>2</sub> were taken in 6 L evacuated Silonite canisters (ENTECH, type: PN: 29- 10622) that were then filled to ambient pressure. Sampling was done through a Mg(ClO<sub>4</sub>)<sub>2</sub> dryer and a filter, and the flow into the canisters was regulated using a manual flow controller. The dryer was changed after every two samples. At the start of each experiment, two canister samples were collected from air<sub>in</sub>, and their average mole fraction and isotope values (Table 1) were used to characterize the incoming air. At each new light setting, and after photosynthetic gas exchange was stable (as monitored with the QCLS and with the LI-7000), two samples were taken

from air<sub>out</sub>. For PAR > 0, these two samples were treated as duplicates and their average mole fraction and isotope values were used for subsequent analyses. In the dark, the plant was still gradually adjusting over time (e.g. closing its stomata) and therefore, these two air<sub>out</sub> samples were not treated as duplicates and their individual data points are reported.

#### 2.3. Experimental conditions

For all experiments, the chamber was supplied with air mixtures with [COS] = 2300–2400 pmol mol<sup>-1</sup>, and [CO<sub>2</sub>] = 430–440 µmol mol<sup>-1</sup> at a flow rate of 8.1 L min<sup>-1</sup>, giving an air residence time of around 1.5–2 min. Temperature in the chamber was 24.6–25.0 °C in sunflower experiments and 25.7–25.9 °C in papyrus experiments, chosen to obtain sufficient COS uptake flux (for isotope analysis) while avoiding condensation of water vapor in the system. Light intensity was sequentially set to PAR = 400, 600, 200, and 0 µmol m<sup>-2</sup> s<sup>-1</sup>, allowing time after each light setting for plant adjustment, uptake flux stabilization and subsequent isotope sampling. Measurements at PAR 600 µmol m<sup>-2</sup> s<sup>-1</sup> were not performed with the papyrus due to time constrains. For the dark measurements, chamber light was switched off and the chamber was covered with a blanket.

#### 2.4. Uptake flux calculations

Both CO<sub>2</sub> and COS net uptake fluxes ( $A^s$  in pmol m<sup>-2</sup>s<sup>-1</sup> and  $A^c$  in µmol m<sup>-2</sup>s<sup>-1</sup>) were calculated using Eq. (5) (which shows the calculation for COS):

$$A^{s} = \frac{u_{e}}{S} \left( C_{e}^{s} - C_{a}^{s} \frac{1 - w_{e}}{1 - w_{a}} \right), \tag{5}$$

where  $u_e$  is the molar flow of air entering the chamber (mol air s<sup>-1</sup>), S is the leaf area (m<sup>2</sup>), and  $w_e$  and  $w_a$  (mol of H<sub>2</sub>O mol air<sup>-1</sup>) are the mole fractions of water vapor in air<sub>in</sub> and air<sub>out</sub>,  $C_e^S$  and  $C_a^S$  (pmol COS mol air<sup>-1</sup>) are the [COS] in air<sub>in</sub> and air<sub>out</sub>, respectively.

The uncertainties of the uptake fluxes were calculated by propagating the uncertainties of the in- and outgoing air mole fraction measurements. In the case of the mole fraction measurements by the QCLS, the  $1\sigma$  uncertainties were obtained measuring air<sub>in</sub> or air<sub>out</sub> during 15 minutes.

As a consistency check, we also calculated the uptake fluxes using the CO<sub>2</sub> and COS mole fractions determined with the mass spectrometer in the canister samples. Comparison of fluxes determined by both methods lead to the exclusion of two samples because of suspected contamination (see Fig. A1 in Appendix A). QCLS COS and CO<sub>2</sub> fluxes, excluding these two samples, were used in subsequent analyses.

From the CO<sub>2</sub> fluxes, the water vapor fluxes obtained from the LI-7000 analyzer and the leaf temperature, we calculated  $C_i^C/C_a^C$  using the gas exchange calculations by Farquhar et al. (1980) (details in Appendix B). The leaf internal COS mole fraction,  $C_i^S$ , was calculated using Eqs. (6) and (7), including a ternary correction:

$$C_i^S = \frac{\left(g_t^S - \frac{E}{2}\right)C_a^S - A^S}{g_t^S + \frac{E}{2}},\tag{6}$$

where  $g_t^s$  is the total leaf conductance to COS from ambient air to the internal leaf space  $(C_i^s)$  (Eq. (7)).

$$g_t^s = \frac{1}{\frac{1.94}{g_s^w} + \frac{1.56}{g_h^w}} \tag{7}$$

Here,  $g_b^w$  is the boundary layer conductance to water, which was assumed infinite, as the chamber fans created well-mixed air. The coefficients 1.94 and 1.56 (mol H<sub>2</sub>O mol COS<sup>-1</sup>) are the ratios of diffusivities of COS to water vapor in air and in the boundary layer, respectively (Fuller et al., 1966; Farquhar & Lloyd, 1993). Equations (6) and (7) assume that the leaf internal spaces are saturated with water vapor. This assumption has been questioned, particularly under high avaporative demands (Cernusak et al., 2018; Cernusak et al., 2024), which were not the conditions during our experiments. Further details on gas exchange calculations are presented in Appendix B.

From the CO<sup>34</sup>S isotope discrimination values ( $^{34}\Delta$ , Eq. (4)), we estimated the COS mole fraction in the mesophyll cell ( $C_m^S$ ), using Eq. (8).

$$C_m^s \cong \frac{C_a^s(\Delta^{34}S - a_b) + C_s^s(a_b - a_s) + C_i^s(a_s - a_m)}{h - a_m},$$
 (8)

where the diffusion fractionation components of  $\bar{a}$  were split into fractionation occurring during boundary layer diffusion ( $a_b = 3.5 \,\%_0$ ), stomatal diffusion ( $a_s = 5.2 \,\%_0$ ) and mesophyll diffusion ( $a_m = 0.5 \,\%_0$ ).  $C_s^s$  is the COS mole fraction at the leaf surface, calculated using Eq. (B14), assuming infinite  $g_b^w$ , and h (=15  $\%_0$ ) is the fractionation occurring during COS hydrolysis by CA (Eq. (4)). The values for all these fractionation factors are from Davidson et al. (2022).

Using a *big leaf* approach, we applied Eqs. (6) to (8) to entire plants excluding roots (sunflower) or several leaves (*papyrus*). This approach assumes that the entire canopy behaves as a single unshaded leaf. In reality, gradients in light or temperature occur within the canopy, but those should have been minor in our experiment that used small plants in a well-mixed chamber. Additionally, given the precision at which the COS isotope exchange can currently be determined, we deemed it unnecessary to go beyond the *big leaf* approach.

#### 2.5. Isotope ratio measurements

COS and CO<sub>2</sub> isotope ratios in the canister samples were determined using isotope ratio mass spectrometry (IRMS) at Utrecht University. Before measurement, the sample canisters' pressure was increased by adding COS-free zero air, as the extraction system needs overpressure. The  $\delta^{34}$ S in COS was determined according to the methods described in Baartman et al. (2022) but using a new Delta V Plus mass spectrometer, which was specifically customized to measure COS isotope ratios with improved performance (Thermo Fisher Scientific, USA). The continuous-flow GC-IRMS system measures the S<sup>+</sup> fragment ions generated in the IRMS ion source by the electron-impact fragmentation of COS. The isotope ratios were calculated relative to our laboratory standard, which is a 50 L cylinder, filled with outside air and spiked with COS to approximately 800 pmol mol<sup>-1</sup> COS. This lab standard was calibrated against the Vienna Canyon Diablo Troilite (VCDT) international sulfur isotope standard (see Baartman et al., 2022 for a detailed description of the COS isotope measurement system). The typical reproducibility error for  $\delta^{34}$ S in COS was 0.4 % and the typical uncertainty for a single sample measurement with ambient COS mole fraction was 0.9 % (Baartman et al., 2022).

The  $\delta^{13}C$  and  $\delta^{18}O$  in  $CO_2$  were measured using a separate continuous flow IRMS system, initially developed for measuring CO isotopologues (Pathirana et al. 2015), and later modified to measure  $CO_2$  isotopologues. A laboratory reference air cylinder with known isotopic composition was used for calibration (Brenninkmeijer, 1993). Typical precision was better than 0.2 % for both  $\delta^{13}C$  and  $\delta^{18}O$ . Values are reported on the Vienna Pee Dee Belemnite (VPDB) ( $\delta^{13}C$ ) and Vienna Standard Mean Ocean Water (VSMOW) ( $\delta^{18}O$ ) scales.

# 2.6. Isotope discrimination calculations

Observed isotope discrimination (‰) was calculated using Eqs. (9) and (10) (Evans et al. 1986):

$$\Delta = \frac{\xi(\delta_a - \delta_e)}{1000 + \delta_a - \xi(\delta_e - \delta_a)},\tag{9}$$

where  $\delta_e$  and  $\delta_a$  are the isotope compositions of the gas entering and leaving the chamber, respectively, for the gas of interest ( $\delta^{13}$ C,  $\delta^{18}$ O in CO<sub>2</sub>, or  $\delta^{34}$ S in COS).  $\xi$  is calculated as:

$$\xi = \frac{C_e}{C_e - C_a},\tag{10}$$

where  $C_e$  and  $C_a$  are the mole fractions (CO<sub>2</sub> or COS), entering and leaving the chamber, respectively. The errors on the measured mole fractions and isotope ratios were propagated to the isotope discrimination values ( $\Delta$ ); details are provided in the supplementary material.

295

#### 3. Results and Discussion

#### 3.1. COS and CO<sub>2</sub> uptake fluxes

In experiments with both plant species there was a net uptake of COS under all light conditions, including dark (Fig. 3b). Mean COS uptake fluxes in the light were  $73.3 \pm 1.5$  pmol m<sup>-2</sup> s<sup>-1</sup> and  $107.3 \pm 1.5$  pmol m<sup>-2</sup> s<sup>-1</sup> for sunflower and papyrus, respectively, and uptake fluxes did not vary strongly for different light conditions. Note that samples in the dark were taken sequentially, when plant performace was still adjusting.

Previously reported COS uptake fluxes at the ecosystem scale usually range between 30 and 60 pmol m<sup>-2</sup> s<sup>-1</sup> (Cho et al., 2023; Kooijmans et al., 2017; Commane et al., 2015; Billesbach et al., 2014), with some higher reported uptake fluxes around 80 to 100 pmol m<sup>-2</sup> s<sup>-1</sup> (Asaf et al., 2013; Spielmann et al., 2023). Berkelhammer et al. (2020) reported maximum mid-day ecosystem-scale COS uptake fluxes of up to 100 pmol m<sup>-2</sup> s<sup>-1</sup> for a maize field (C<sub>4</sub>) during July. Those values were higher than the mid-day fluxes obtained from a prairie (C<sub>3</sub> and C<sub>4</sub> species), being around 50 pmol m<sup>-2</sup> s<sup>-1</sup> (July – August). However, Stimler et al. (2011) measured COS fluxes of only around 30 pmol m<sup>-2</sup> s<sup>-1</sup> under similar light intensity, in leaf cuvette experiments. Thus, our measured COS uptake fluxes are at the high end of the spectrum.

315

Stomatal conductance to water vapor in sunflower ranged from 0.25 to 0.35 mol m<sup>-2</sup> s<sup>-1</sup> under light conditions and decreased to 0.15 mol m<sup>-2</sup> s<sup>-1</sup> in the dark (Table 2). In papyrus, stomatal conductance was slightly higher in the light, ranging between 0.27 and 0.39 mol m<sup>-2</sup> s<sup>-1</sup>. In the dark, stomatal conductance for papyrus dropped substantially to 0.09 mol m<sup>-2</sup> s<sup>-1</sup> during the first sampling and further to 0.04 mol m<sup>-2</sup> s<sup>-1</sup> during the second. This is reflected in the lower COS assimilation for papyrus in the dark compared to sunflower (see Fig. 3 and Table 2).

Overall, our observed stomatal conductance values are at the upper end of the previously reported ranges. For example, Stimler et al. (2011) reported g<sub>s</sub> values of up to approximately 0.17 mol m<sup>-2</sup> s<sup>-1</sup>, while Berkelhammer (2020) found maximum g<sub>s</sub> values of around 0.22 mol m<sup>-2</sup> s<sup>-1</sup> for maize (C<sub>4</sub>) and 0.12 for a prairie field (C<sub>3</sub> and C<sub>4</sub>). Miner & Bauerle (2017) did find unusually high stomatal conductance values for sunflowers of up to 1.2, with a high inter-plant variability and Howard & Donovan (2007) reported nighttime g<sub>s</sub> values of 0.023-0.225 for well-watered sunflowers. These elevated g<sub>s</sub> values in our experiments likely explain the relatively high and stable COS fluxes for PAR > 0. Moreover, the non-zero g<sub>s</sub> values under PAR = 0 support the continued COS uptake in the dark, particularly for sunflower (Figure 3b). As hydrolysis of COS, catalyzed by CA, is a light-independent reaction, COS assimilation can continue as long as the stomata are open (Protoschill-Krebs et al., 1996).

The small increase in  $C_i^S/C_a^S$  values (Table 2) with increasing PAR also suggests that stomata were sufficiently open to sustain stable COS uptake fluxes, even in low-light conditions. In plant experiments conducted with elevated COS mole fractions (1.5 nmol mol<sup>-1</sup>), Stimler et al. (2010) reported similar  $C_i^S/C_a^S$  values around 0.6, corresponding to COS uptake fluxes around 100 pmol m<sup>-2</sup> s<sup>-1</sup> and g<sub>s</sub> of 0.5 mol m<sup>-2</sup> s<sup>-1</sup>. Thus, the higher than usual  $C_i^S/C_a^S$  and potentially the higher stomatal conducance in our experiments may be attributable to the elevated COS mole fractions in our chamber. These elevated COS mole fractions were necessary for obtaining precise measurements of COS isotope discrimination.

Figure 3. a:  $A^C$  (CO<sub>2</sub> uptake flux, in  $\mu$ mol  $m^{-2}$  s<sup>-1</sup>), b:  $A^S$  (COS uptake flux, in  $\mu$ mol  $m^{-2}$  s<sup>-1</sup>) and c: LRU versus PAR ( $\mu$ mol  $m^{-2}$  s<sup>-1</sup>), for sunflower (orange stars) and papyrus (green circles). Flux values for PAR > 0 are means  $\pm 1$  standard error (SE) (n = 2), where 1 SE was obtained using error propagation (see supplementary materials), flux values for PAR = 0 reflect individual measurements. Only positive LRU values are shown. LRU was negative for PAR = 0 (see Table 2). Errors are only displayed when larger than the symbols.

Both sunflower and papyrus respired  $CO_2$  in the dark and photosynthesyzed in the light, at a net rate that increased with PAR (Fig. 3a). Mean  $CO_2$  uptake fluxes in light conditions were  $6.7 \pm 1.7$  µmol m<sup>-2</sup> s<sup>-1</sup> for sunflower and  $11.7 \pm 2.2$  µmol m<sup>-2</sup> s<sup>-1</sup> for papyrus (Fig. 3a). These photosynthesis rates match that of sunflowers of Tezera et al. (2008) under their low-light condition experiments (in the least drought-exposed conditions).

At all light intensities (PAR > 0), CO<sub>2</sub> uptake rates were larger in papyrus than in sunflower, matching expectations for C<sub>4</sub> vs. C<sub>3</sub> photosynthesis (Farquhar & Lloyd, 1993). Our measurements can be classified as relatively low-light, because although the PAR measured at the top of the chamber was 400  $\mu$ mol m<sup>-2</sup> s<sup>-1</sup> at the highest setting for the C<sub>4</sub> experiments, there was likely light attenuation across the plant canopy. The photosynthesis rates for papyrus are comparable with previous measurements, conducted under low-light conditions. Ubierna et al., (2013) measured CO<sub>2</sub> assimilation rates of around 10  $\mu$ mol m<sup>-2</sup> s<sup>-1</sup> at PAR = 500  $\mu$ mol m<sup>-2</sup> s<sup>-1</sup> in three C<sub>4</sub> species, *Zea mays, Miscanthus* 

x giganteus and Flaveria bidentis, under varying light conditions between 0 and 2000  $\mu$ mol m<sup>-2</sup> s<sup>-1</sup>. Their results are similar to our measured CO<sub>2</sub> uptake fluxes of between 9.4  $\mu$ mol m<sup>-2</sup> s<sup>-1</sup> (200 PAR) and 14.0  $\mu$ mol m<sup>-2</sup> s<sup>-1</sup> (400 PAR).

At PAR = 600  $\mu$ mol m<sup>-2</sup> s<sup>-1</sup>, LRU (Eq. (1)) was 2.3 ± 0.08 for sunflower and at PAR = 400  $\mu$ mol m<sup>-2</sup> s<sup>-1</sup>, LRU values were 3.1 ± 0.11 and 1.7 ± 0.06 for sunflower and papyrus, respectively (see Table 2 and Fig. 3). As PAR decreased to 200  $\mu$ mol m<sup>-2</sup> s<sup>-1</sup>, LRU increased to 5.2 ± 0.16 for sunflower and 3.0 ± 0.11 for papyrus. The increase in LRU at low light was due to a decrease in CO<sub>2</sub> uptake fluxes while the COS uptake remained roughly constant. In the dark, LRU values were negative, up to -16.0 for sunflower, as COS uptake by the plant continued while CO<sub>2</sub> was being respired. Our LRU values are higher than those found by Stimler et al. (2011) and higher than the usually reported median LRU values of 1.7 (n = 53) for C<sub>3</sub> species and 1.2 (n = 4) for C<sub>4</sub> (Whelan et al., 2018), which may be due to our relatively low-light experiments. Still, previously reported LRU values display a wide range of values of between 0.7 and 6.2, and Stimler et al. (2011) also reported a higher LRU for C<sub>4</sub> compared to C<sub>3</sub>. Furthermore, recent research has shown that LRU can differ across species and vary with environmental conditions, especially light availability and VPD (Kooijmans et al., 2019; Spielmann et al., 2023; Sun et al., 2022). The exact mechanism for this varying LRU is still not completely understood (Whelan et al., 2018; Wohlfahrt et al., 2023).

Our slightly high LRU values could also be due to the higher than ambient COS mole fractions (of around 2 nmol mol<sup>-1</sup>) that the plants were exposed to during our experiments. Davidson et al. (2022) reported LRU values or 0.7 and 1.7 for C<sub>3</sub> and C<sub>4</sub>, respectively for experiment with ambient COS mole fractions, and LRU values of 2.4 and 1.0 for C<sub>3</sub> and C<sub>4</sub> for plants exposed to 2900 µmol mol<sup>-1</sup> CO<sub>2</sub> and 3.4 nmol mol<sup>-1</sup> COS (see Appendix C). Thus, exposure to higher COS mole fractions could have influenced LRU, however, in the experiments by Davidson et al (2022), not only the COS but also the elevated CO<sub>2</sub> mole fractions could have affected the LRU (Sun et al., 2022).

Figure 4 shows the CO<sub>2</sub> uptake flux ( $\mu$ mol m<sup>-2</sup> s<sup>-1</sup>) plotted against ratio of the CO<sub>2</sub> mole fractions in the intercellular space versus the ambient (Table 2) ( $C_i^C/C_a^C$ ). The  $C_i^C/C_a^C$  ratio increases with decreasing CO<sub>2</sub> uptake flux for both species and the differences in CO<sub>2</sub> uptake flux between C<sub>3</sub> and C<sub>4</sub> are consistent with the results presented by Stimler et al. (2011). Our measured  $C_i^C/C_a^C$  for sunflower compares well with previous values for sunflower of 0.8 found by Tezara et al. (2008). The  $C_i^C/C_a^C$  for papyrus is high for a C<sub>4</sub> species, for which values usually range around 0.4, but could again be explained by the low-light conditions, as previously observed by Ubierna et al., (2013). The higher than usual  $C_i^C/C_a^C$  could also be explained by the fact that we measured entire plants, of which some leaves were partly shaded.

Figure 4.  $C_i^C/C_a^C$  plotted against  $A^C$  (CO<sub>2</sub> uptake flux in  $\mu$ mol  $m^{-2}$   $s^{-1}$ ), for sunflower (stars) and papyrus (circles). Colors indicate PAR levels ( $\mu$ mol  $m^{-2}$   $s^{-1}$ ). Data for PAR = 0 are not included because the plants were respiring during dark conditions.

# 3.2 CO<sup>34</sup>S discrimination

Table 2 shows the isotopic discrimination for COS ( $^{34}\Delta$ ) and CO<sub>2</sub> ( $^{13}\Delta$ ,  $^{18}\Delta$ ), and accompanying data for the different light treatments. In contrast to the CO<sub>2</sub> isotope discrimination (Sect. 3.3),  $^{34}\Delta$  did not show a trend with COS uptake flux nor with PAR (Fig. 5),  $C_i^S/C_a^S$  (Fig. 6), or a difference between the species. The average  $^{34}\Delta$  values in light conditions (PAR > 0) were  $3.4 \pm 1.0$  (SEM) % for sunflower and  $2.6 \pm 1.0$  (SEM) % for papyrus (see Table 2). For sunflower in dark conditions, we found a  $^{34}\Delta$  of  $4.7 \pm 1.5$  % for the first sample and  $1.3 \pm 1.6$  % for the second sample. The COS uptake flux for papyrus in dark conditions decreased drastically, to the point that  $^{34}\Delta$  could no longer be estimated with confidence (see Fig. 3).

Table 2. Photosynthetic discrimination (mean  $\pm$  1 SE, n=2), COS and CO<sub>2</sub> uptake fluxes ( $A^S$  and  $A^C$ ), LRU, stomatal conducance to water vapor ( $g_{sw}$ ), total conductance to  $COS(g_t^s)$ , leaf internal vs. ambient mole fraction ratios for  $COS(C_i^s/C_a^s)$  and  $CO_2(C_i^c/C_a^c)$ , mesophyll vs. ambient COS mole fraction  $(C_m^s/C_a^s)$ for sunflower and papyrus, for each PAR level. The uncertainties were calculated as the standard error of the mean (SEM) and the student's t-distribution, with 60% confidence interval and 1 (=n-1) degree of freedom. Uncertainties where n=1 are the propagated measurement uncertainties. Values without stated uncertainty are single sample measurements (in the case of isotope discrimination values) or have an uncertainty smaller than 0.01 (in the case of  $C_i^S/C_a^S$  and  $C_m^S/C_a^S$ ).  $A^S$  at PAR = 0 for papyrus was too small for calculating  $^{34}\Delta$ . The samples taken in the dark were not seen as duplicates as the plant was still adjusting to

the dark conditions between sampling, and two values for PAR = 0 are given for each species.

| Plant                  | PAR                                     | Number        | $^{34}\Delta$ | $^{13}\Delta$ | <sup>18</sup> ∆ (‰) | $A^{S}$                                    | $A^{C}$                                               | LRU        | $\boldsymbol{g}_{sw}$                     | $\boldsymbol{g}_t^s$                      | $oldsymbol{g}^c_t$                        | $C_i^S/C_a^S$ | $C_m^S/C_a^S$                                    | $C_i^C/C_a^C$                         |
|------------------------|-----------------------------------------|---------------|---------------|---------------|---------------------|--------------------------------------------|-------------------------------------------------------|------------|-------------------------------------------|-------------------------------------------|-------------------------------------------|---------------|--------------------------------------------------|---------------------------------------|
|                        | (μmol m <sup>-2</sup> s <sup>-1</sup> ) | of<br>samples | (‰)           | (‰)           |                     | (pmol<br>m <sup>-2</sup> s <sup>-1</sup> ) | (μmol<br>m <sup>-2</sup> s <sup>-1</sup> )            |            | (mol<br>m <sup>-2</sup> s <sup>-1</sup> ) | (mol<br>m <sup>-2</sup> s <sup>-1</sup> ) | (mol<br>m <sup>-2</sup> s <sup>-1</sup> ) |               |                                                  | , , , , , , , , , , , , , , , , , , , |
| ~ a                    | 200                                     |               |               | 22.1          | 110 -               |                                            |                                                       |            | 0.05                                      | 0.10                                      | 0.16                                      | 0.50          | 0.11                                             | 0.04                                  |
| Sunflower              | 200                                     | 2             | 3.6 ± 1.6     | 32.4<br>± 1.1 | $148.7 \pm 0.7$     | 72.1 ± 1.9                                 | $\begin{array}{ccc} 4.42 & \pm \\ 0.02 & \end{array}$ | 5.2 ± 0.16 | 0.25                                      | 0.13                                      | 0.16                                      | 0.50          | $\begin{array}{c} 0.11 \\ (n=1)^{c} \end{array}$ | 0.91                                  |
| Sunflower              | 400                                     | 1             | 3.7 ± 2.3     | 24.9<br>± 1.5 | 83.6 ± 1.5          | 72.3 ± 2.2                                 | 6.86 ± 0.02                                           | 3.1 ± 0.11 | 0.26                                      | 0.14                                      | 0.16                                      | 0.52          | 0.07                                             | 0.86                                  |
| Sunflower              | 600                                     | 2             | 2.8 ± 1.7     | 23.6<br>± 1.2 | 63.8 ± 0.9          | 74.9 ± 2.1                                 | 8.81 ± 0.02                                           | 2.3 ± 0.08 | 0.35                                      | 0.18                                      | 0.22                                      | 0.62          | 0.04                                             | 0.87                                  |
| Sunflower <sup>a</sup> | 0                                       | 1             | 4.7 ± 1.5     | -             | -                   | 59.9 ± 1.9                                 | -                                                     | -          | 0.16                                      | 0.08                                      | 0.10                                      | 0.45          | -                                                | -                                     |
| Sunflower <sup>a</sup> | 0                                       | 1             | 1.3 ± 1.3     | -             | -                   | 58.0 ± 1.8                                 | -                                                     | -          | 0.15                                      | 0.08                                      | 0.09                                      | 0.45          | -                                                | -                                     |
| Papyrus                | 200                                     | 1             | 2.5 ± 1.6     | 21.8<br>± 1.5 | 79.4 ± 1.5          | 108.6 ± 3.5                                | 9.36 ± 0.04                                           | 3.0 ± 0.11 | 0.27                                      | 0.14                                      | 0.17                                      | 0.39          | 0.05                                             | 0.82                                  |
| Papyrus                | 400                                     | 2             | 2.6 ± 1.4     | 18.9<br>± 3.4 | 49.4 ± 0.4          | 105.9 ± 3.1                                | 14.01 ± 0.08                                          | 1.7 ± 0.06 | 0.39                                      | 0.20                                      | 0.24                                      | 0.58          | $0.03$ $(n = 1)^{c}$                             | 0.79                                  |
| Papyrus <sup>b</sup>   | 0                                       | 1             | -             | -             | -                   | 33.8 ± 3.6                                 | -                                                     | -          | 0.09                                      | 0.05                                      | 0.05                                      | 0.60          | -                                                |                                       |
| Papyrus <sup>b</sup>   | 0                                       | 1             | -             | -             | -                   | 15.3 ± 3.1                                 | -                                                     | -          | 0.04                                      | 0.02                                      | 0.03                                      | 0.66          | -                                                |                                       |

<sup>&</sup>lt;sup>a</sup>There was no uptake of  $CO_2$  at PAR = 0

<sup>&</sup>lt;sup>b</sup>There was no uptake of CO<sub>2</sub> at PAR = 0 and not sufficient COS uptake to calculate  $^{34}\Delta$ 

 $<sup>{}^{</sup>c}C_{m}^{S}/C_{a}^{S}$  only obtained from one sample as the calulations for the other sample yielded negative (unrealistic) values for  $C_{m}^{S}$ 

Figure 5. Plant COS isotope discrimination ( $^{34}\Delta$ ) plotted against  $A^S$  (COS uptake flux in pmol m $^{-2}$  s $^{-1}$ ) for sunflower (stars) and papyrus (circles). Colors indicate PAR levels (µmol m $^{-2}$  s $^{-1}$ ). Samples for PAR = 0 are only shown for sunflower as  $A^S$  for papyrus (PAR = 0) was too low to calculate  $^{34}\Delta$  with meaningful precision.

Figure 6. Plant COS isotope discrimination ( $^{34}\Delta$ ) against the ratio of internal versus  $C_i^S/C_a^S$ , for sunflower (stars) and papyrus (circles). Colors indicate PAR levels (µmol m<sup>-2</sup> s<sup>-1</sup>). Samples for PAR = 0 are only shown for sunflower as  $A^s$  for papyrus (PAR = 0) was too low to calculate  $^{34}\Delta$  with meaningful precision.

To further investigate this lack of variability in  $^{34}\Delta$ , we examined the variability in  $C_i^S/C_a^S$  and  $C_m^S/C_a^S$  as a function of PAR (Table 2). We observed a slight increase of  $C_i^S/C_a^S$  with PAR that could be explained by an increase in  $g_s$  with available light. Observed COS isotope discrimination also depends on  $C_m^S/C_a^S$ , the ratio of COS mole fractions in the mesophyll cell and the ambient air (see Eq. (4)). This ratio was relatively stable at low values around 0.03–0.07 (Table 2) over the various PAR levels and did not differ substantially between sunflower and papyrus, except for one sunflower sample (PAR = 200) yielding a  $C_m^S/C_a^S = 0.11$ . This lack in variability in  $C_m^S/C_a^S$  might explain the absence in variability in  $C_m^S/C_a^S$  does entail several assumptions (see Eq. (B16) – B(19) in Appendix B), and thus, the results should not be over interpreted.

Comparing our  $^{34}\Delta$  to previous studies, Angert et al. (2019) estimated a value for  $^{34}\Delta$  during COS plant uptake of around 5 ‰ (based on binary diffusion theory), and experiments presented by Davidson et al. (2021) and Davidson et al. (2022) yielded  $^{34}\Delta$  values of  $1.6 \pm 0.1$  ‰ for  $C_3$  and  $5.4 \pm 0.5$  ‰ for  $C_4$  species. Our results differ from these measurements, as we did not find statistically different  $^{34}\Delta$  values between our  $C_3$  and  $C_4$  species. However, the range for  $^{34}\Delta$  that we measured in sunflower of  $2.8 \pm 1.7$  ‰ to  $3.7 \pm 2.3$  ‰ (average  $3.3 \pm 1.0$  (SEM) ‰) is in the same range as the  $C_3$   $^{34}\Delta$  found by Davidson et al. (2021; 2022) and the theoretical estimate of Angert et al. (2019). This is reassuring, given that different measurement techniques were used for both the plant experiments (flow-through chamber compared to closed-chamber) and the isotope ratio measurements.

The benefit of using a flow-through system is that stable environmental conditions inside the chamber can be maintained during the experiment. In contrast, in a closed chamber, CO<sub>2</sub> and COS mole fractions will decrease due to plant uptake, which can be problematic when the experiment runs over long periods of time. Furthermore, transpiration by the plant will increase the water vapor mole fraction in the chamber, which might affect stomatal opening and therefore also the isotope fractionation.

# 3.3 CO<sub>2</sub> isotope discrimination

### 3.3.1 <sup>13</sup>CO<sub>2</sub> discrimination

In both sunflower and papyrus,  $^{13}\Delta$  increased as the CO<sub>2</sub> uptake flux decreased, with decreasing PAR (Fig. 7). Average  $^{13}\Delta$  in sunflower was between 23.6 ± 1.2 and 32.4 ± 1.1 ‰ (Table 2), which is within the range of values expected for C<sub>3</sub> photosynthesis (Farquhar et al. 1982, Kohn 2010, Cernusak et al. 2013, Wingate et al., 2007). However, in papyrus,  $^{13}\Delta$  was between 18.9 ± 3.4 and 21.8 ± 1.5 ‰; much larger than the expected 3 to 6 ‰ for C<sub>4</sub> species operating at optimal conditions (Farquhar et al 1983; Cerling et al. 1997; Kubásek et al., 2013; Ellsworth and Cousins, 2016; Eggels et al., 2021). As previously explained, our measurements were performed at low light intensities (PAR ≤ 400 µmol m<sup>-2</sup> s<sup>-1</sup>), which resulted in moderately low photosynthetic rates (9.3-14.0 µmol m<sup>-2</sup> s<sup>-1</sup>). In C<sub>4</sub> species,  $^{13}\Delta$  has been shown to increase at low light to values as large as 8-17‰, when PAR = 50-125 µmol m<sup>-2</sup>s<sup>-1</sup> (Ubierna et al. 2013, Pengelly et al. 2010, Kromdijk et al. 2010) and photosynthetic rates were small (<5 µmol m<sup>-2</sup>s<sup>-1</sup>). Our  $^{13}\Delta$  values for papyrus are still larger than these previous reports at low irradiance, suggesting that processes other than photosynthesis might have affected the measurements. Upward transport of water dissolved CO<sub>2</sub> in the transpiration stream has been shown in tree stems (Aubrey and Teskey, 2009; Bloemen et al. 2013) and in papyrus culms (Li and

Jones, 1995). We measured detached papyrus leaves submerged in water. This setting could have facilitated the transport of water dissolved  $CO_2$  into the leaf chamber, particularly because papyrus leaves have numerous vascular bundles surrounded by large air cavities (Plowman, 1906). Water dissolved  $CO_2$  would presumably have near-ambient air  $\delta^{13}C$  values – enriched compared to tank  $CO_2$  supplied to the chamber air –, and therefore if released in the plant chamber would artefactually increase  $^{13}\Delta$ .

Figure 7. Variation of photosynthetic discrimination against  $^{13}CO_2(^{13}\Delta$ , panel a) and  $CO^{18}O(^{18}\Delta$ , panel b) as a function of  $A^C(CO_2)$  uptake flux in  $\mu$ mol  $m^{-2}$  s $^{-1}$ ) for sunflower (stars) and papyrus (circles). Colors indicate PAR levels ( $\mu$ mol  $m^{-2}$  s $^{-1}$ ). Data for PAR = 0 are not included because the plants were respiring in during dark conditions.

#### 3.3.2 C<sup>16</sup>O<sup>18</sup>O discrimination

From Fig. 7, we observe a negative relationship between apparent  $^{18}\Delta$  and CO<sub>2</sub> uptake flux, similar to  $^{13}\Delta$ . The average  $^{18}\Delta$  values of sunflower range between  $63.8 \pm 0.9$  and  $148.7 \pm 0.7$  ‰ and the average  $^{18}\Delta$  values of papyrus are between  $49.4 \pm 0.4$  and  $79.4 \pm 1.5$  ‰ (Table 2).  $^{18}\Delta$  mostly reflects the exchange of  $^{18}O$  between CO<sub>2</sub> and leaf water (Francey and Tans; Yakir, 1998; Adnew et al., 2020). The lower  $^{18}\Delta$  in C<sub>4</sub> species likely indicates the incomplete equilibrium between CO<sub>2</sub> and leaf water, because of the reduced CA activity in C<sub>4</sub> species compared to most C<sub>3</sub> species (Gillon and Yakir, 2000).

A negative correlation of  $^{18}\Delta$  with CO<sub>2</sub> assimilation and light intensity, as well as lower  $^{18}\Delta$  in C<sub>4</sub> species was also found by Stimler et al. (2011). For their C<sub>3</sub> plants, they found that  $^{18}\Delta$  ranged between 40 and 240 ‰, with the highest values found at the lowest CO<sub>2</sub> uptake fluxes. For C<sub>4</sub> species, Stimler et al. (2011) found an  $^{18}\Delta$  between 10 and 50 ‰. Seibt et al. (2006) also found large variations in  $^{18}\Delta$  during CO<sub>2</sub> uptake by *Picea sitchensis*, and a correlation with PAR. They too measured the largest  $^{18}\Delta$  discrimination at dusk and dawn, when light intensity was lowest.

The relation between the COS uptake flux and  $^{18}\Delta$  can also be analyzed, since both depend on the same diffusion pathway and CA activity (Stimler et al., 2011). Stimler et al. (2011) observed a negative correlation between  $^{18}\Delta$  and COS uptake flux, with a larger change in  $^{18}\Delta$  for C<sub>3</sub> species, compared to C<sub>4</sub>. Figure 8 shows  $^{18}\Delta$  against the COS uptake flux for our data. We do not observe such a correlation between  $^{18}\Delta$  and the uptake COS flux. However, our range in COS uptake flux for each species is small, as we found that the COS uptake flux did not change

significantly with light intensity. In the same range of COS uptake flux data, Stimler et al. (2011) did not find a strong trend in  $^{18}\Delta$  either.

Figure 8.  $^{18}\Delta$  (‰) plotted against A<sup>S</sup> (COS uptake flux in pmol m<sup>-2</sup> s<sup>-1</sup>) for sunflower (C3) and papyrus (C4), where the different symbols and colors indicate the plant types and PAR (µmol m<sup>-2</sup> s<sup>-1</sup>). Data for PAR = 0 are not included because the plants were respiring during dark conditions.

#### 4 Conclusions & perspectives

This study presented measurements of COS and CO<sub>2</sub> plant uptake fluxes and COS ( $^{34}\Delta$ ) and CO<sub>2</sub> ( $^{13}\Delta$  and  $^{18}\Delta$ ) isotope discrimination for sunflower (C<sub>3</sub>) and papyrus (C<sub>4</sub>). The experiments were conducted using a flow-through gas exchange system, which is a new and different method compared to previously reported measurements of COS isotope fractionation during plant uptake (Davidson et al., 2021; 2022). The gas exchange system including the QCLS and LI-7000 instruments ensured stable chamber conditions, which were easy to monitor throughout the experiments.

Our study is the first to combine measurements of both COS and CO<sub>2</sub> plant isotope discrimination, where the CO<sub>2</sub> values provided additional information on the plant's behavior and their responses to environmental variation. CO<sub>2</sub> assimilation increased with increasing PAR level and CO<sub>2</sub> uptake flux was higher for the C<sub>4</sub> than for the C<sub>3</sub> species, both findings being consistent with previous results under similar conditions. However, the moderate to low-light conditions were limiting CO<sub>2</sub> assimilation rate. Corresponding CO<sub>2</sub> isotope discrimination values, <sup>13</sup> $\Delta$  and <sup>18</sup> $\Delta$ , were therefore higher than those normally exhibited by planst at full photosynthetic capacity. CO<sub>2</sub> isotope discrimination as well as  $C_i^C/C_a^C$  were lower in papyrus than in sunflower, consistent with differences between C<sub>3</sub> and C<sub>4</sub> photosynthesis and  $C_i^C/C_a^C$  decreased with light intensity for both species. Therefore, we conclude that both species were behaving normal, albeit not in the most optimal conditions for maximum photosynthetic CO<sub>2</sub> assimilation.

In contrast to photosynthesis, COS assimilation did not vary strongly with light intensity, which is to be expected when stomatal conductance is sufficiently large to maintain a steady COS supply to the mesophyll cell, as

the hydrolysis reaction catalyzed by CA is light-independent. The observed COS uptake flux was lower during the dark experiments, but not zero, consistent with residual stomatal opening. Our measurements also showed a constant  $^{34}\Delta$  across different light settings, which can be explained by the rather constant  $C_i^S/C_a^S$  and  $C_m^S/C_a^S$  values. Surprisingly,  $^{34}\Delta$  also did not differ significantly between papyrus and sunflower, whereas previous measurements (Davidson et al., 2022) reported higher  $^{34}S$  isotope discrimination for C<sub>4</sub> species. Nevertheless, our values for  $^{34}\Delta$  are close to the previously reported values by Davidson et al. (2022), despite using a different experimental set-up and a different way to calculate the isotopic discrimination (Evans et al., 1986).

For future studies, we recommend to use representative C<sub>3</sub> and C<sub>4</sub> plant species to characterize isotope discrimination more broadly. In our study, papyrus was selected due to its availability and large leaf area, which enabled sufficient COS uptake fluxes for isotope analysis at the required precision. However, we acknowledge that papyrus, along with the environmental conditions during our measurements, may not be broadly representative of typical C<sub>4</sub> species. Future work should aim to include a wider range of species and ideally those that are ecologically abundant and physiologically representative of the C<sub>3</sub> and C<sub>4</sub> photosynthetic pathways.

We furthermore recommend to perform experiments under environmental conditions closer to natural field conditions, in particular using higher PAR than in our experiments. However, measuring at high PAR in a plant chamber, while maintaining a sufficient COS mole fraction difference between in- and outgoing air to quantify COS isotope discrimination may introduce technical challenges, especially related to water condensation on chamber walls and sampling lines, which will need to be overcome.

Aditionally, the influence of soil water availability, VPD, and nutrient availability on COS isotope discrimination remains unexplored. Investigating these environmental variables may yield insights into mesophyll conductance and its influence on the LRU.

Finally, we recommend future studies to directly measure the isotope discrimination occurring during the CA-catalyzed hydrolysis of COS. Precisely quantifying the CA discrimination factor, h, as defined in Eq. (4), would provide a critical constraint on possible values for total observed isotope discrimination across different plant species. This would be beneficial for upscaling the isotope signatures to the global scale. Furthermore, better constraining h would enable more accurate estimations of CA activity, thereby improving our understanding of the physiological processes underlying plant COS assimilation.

# **Appendices**

# **Appendix A: Supplementary figures**

Figure A1. CO<sub>2</sub> and COS fluxes in μmol m<sup>-2</sup> s<sup>-1</sup> and pmol m<sup>-2</sup> s<sup>-1</sup>, respectively, calculated from the discrete samples that were analyzed on the mass spectrometer, plotted against the fluxes that were calculated from the online QCLS measurements. Uncertainty bars are ± 1σ, obtained using error propagation of the measurement errors on all the components used during the flux calculations (see supplementary materials). The errors are only depicted when they are larger than the symbols. The stars symbols are the sunflower data, and the circles are the papyrus data. The different color shadings indicate the varying PAR levels in μmol m<sup>-2</sup> s<sup>-1</sup>. The black dashed line shows the one-to-one line, for reference. The two samples that clearly fall off the line in the CO<sub>2</sub> plot were excluded from both the CO<sub>2</sub> and COS dataset, as these sample canisters had possibly leaked or were contaminated with air other than the plant chamber air.

Figure A2. Pictures of the plant chamber, with sunflower (left) and papyrus leaves (right) inside. The chamber consists of two cylinders, connected to each other and to the upper and lower panels with Terostat RB VII. The plant pot and soil are kept outside of the chamber and the chamber is sealed onto the stem with Terostat as well. The black wires are automated (computer controlled) heating wires, ensuring constant temperature around the chamber.

#### Appendix B: Gas exchange calculations for CO2 and COS

We detail gas exchange equations of von Caemmerer and Farquhar (1981) for CO<sub>2</sub> and adapt this theory to derive gas exchange parameters for COS. For assimilation rates and mixing ratios we adopt a nomenclature where the superscript c refers to CO<sub>2</sub> and s to COS. For conductances the subscript represents the molecule of interest (w – water, c – CO<sub>2</sub>, s – COS) and the superscript the type of conductance (t – total, b – boundary layer, s – stomata).

 $\underline{\text{CO}_2}$  and  $\underline{\text{COS}}$  assimilation rates ( $A^c$ ,  $A^s$ ,  $\mu\text{mol CO}_2$  m<sup>-2</sup> s<sup>-1</sup>,  $A^s$  given by Eq. (5)):

565 
$$A^{c} = \frac{u_{e}}{S} \left( c_{e}^{c} - c_{a}^{c} \frac{1 - w_{e}}{1 - w_{a}} \right), \tag{B1}$$

where  $u_e$  is the molar flow of air entering the chamber (mol air s<sup>-1</sup>), S is the leaf area (m<sup>2</sup>),  $c_e^c$  and  $c_a^c$  (µmol CO<sub>2</sub> mol air<sup>-1</sup>) are the [CO<sub>2</sub>] in the air entering and leaving the chamber, respectively, and  $c_e^s$  and  $c_a^s$  (pmol COS mol air<sup>-1</sup>) are the [COS] in the air entering and leaving the chamber, respectively.

Transpiration rate (mol H<sub>2</sub>O m<sup>-2</sup>s<sup>-1</sup>)

$$E = \frac{u_e}{S} \frac{w_a - w_e}{1 - w_a},\tag{B2}$$

where  $w_e$ ,  $w_a$  (mol of H<sub>2</sub>O mol air<sup>-1</sup>) are the mole fractions of water vapor in the air *entering* the chamber and in the chamber *air* (which equals to the air *out* of the chamber).

Total conductance to water vapor ( $g_w^t$ , mol H<sub>2</sub>O m<sup>2</sup> s<sup>-1</sup>):

$$g_w^t = E \frac{1 - \frac{w_i + w_a}{2}}{w_i - w_a},\tag{B3}$$

where (mol of H<sub>2</sub>O mol air<sup>-1</sup>) is the mole fraction of water vapor *inside* the leaf, which assuming saturation with water vapour at the leaf temperature ( $T_l$ ,  ${}^{\circ}$ C) can be calculated:

$$w_i = \frac{0.61635e^{\frac{17.502T_l}{240.97+T_l}}}{P_a},\tag{B4}$$

where  $P_a$  (kPa) is atmosphere pressure in the chamber.

Stomata conductance to water  $(g_s^w, \text{ mol H}_2\text{O m}^{-2} \text{ s}^{-1})$  is:

$$g_s^w = \frac{1}{\frac{1}{g_t^w} - \frac{1}{g_b^w}},\tag{B5}$$

where  $g_b^w$  is the boundary layer conductance to water, a characteristic of each plant chamber, but often very large in well stirred chambers (a requisite for gas exchange).

Total conductance to  $CO_2$  ( $g_t^c$ , mol  $CO_2$  m<sup>-2</sup> s<sup>-1</sup>) and COS ( $g_t^s$ , mol COS m<sup>-2</sup> s<sup>-1</sup>):

$$g_t^c = \frac{1}{\frac{1.6}{q_t^w} + \frac{1.37}{q_t^w}},\tag{B6}$$

$$g_t^s = \frac{1}{\frac{1.94}{g_s^w} + \frac{1.56}{g_b^w}},\tag{B7}$$

where the coefficient 1.6 and 1.37 (mol H<sub>2</sub>O mol CO<sub>2</sub><sup>-1</sup>) are the ratio of diffusivities of CO<sub>2</sub> to water vapor in air, and in the boundary layer, respectively. The coefficients 1.94 and 1.56 (mol H<sub>2</sub>O mol COS<sup>-1</sup>) are the ratio of diffusivities of COS to water vapor in air, and boundary layer, respectively (Fuller *et al.*, 1966; Farquhar & Lloyd, 1993).

Concentration inside the leaf of CO<sub>2</sub> ( $c_i^c$ , µmol CO<sub>2</sub> mol wet air<sup>-1</sup>) and COS ( $c_i^s$ , pmol COS mol wet air<sup>-1</sup>)

 $A^c$  and  $A^s$  are determined with gas exchange with Eqs. (B1) and (5), and can also be related to the [CO<sub>2</sub>] and [COS] inside the leaf with the equations:

$$A^{c} = g_{t}^{c}(c_{a}^{c} - c_{i}^{c}) - E\frac{c_{a}^{c} + c_{i}^{c}}{2},$$
(B8)

$$A^{s} = g_{t}^{s}(c_{a}^{s} - c_{i}^{s}) - E\frac{c_{a}^{s} + c_{i}^{s}}{2},$$
(B9)

where  $E \frac{c_a^c + c_i^c}{2}$  and  $E \frac{c_a^s + c_i^s}{2}$  are ternary corrections that accounts for the influence of transpiration on the diffusion of CO<sub>2</sub> and COS into the leaf. Solving  $c_i^c$  from Eq. 9 and  $c_i^s$  from Eq. (B9) results in:

$$c_i^c = \frac{\left(g_t^c - \frac{E}{2}\right)c_a^c - A^c}{g_t^c + \frac{E}{2}},\tag{B10}$$

$$c_i^s = \frac{\left(g_t^s - \frac{E}{2}\right)c_a^s - A^s}{g_t^s + \frac{E}{2}}.$$
 (B11)

COS concentration in the mesophyll at the sites of CA ( $c_{m}^{s}$ , pmol COS mol wet air<sup>-1</sup>):

By analogy with the model for photosynthetic discrimination against <sup>13</sup>CO<sub>2</sub> (Farquhar *et al.*, 1982; Farquhar & Cernusak, 2012) discrimination against CO<sup>36</sup>S (‰) during plant uptake can be described:

$$\Delta^{34}S = \frac{1}{1-t}\overline{a_{c_i^s}}\frac{c_a^s - c_i^s}{c_a^s} + \frac{1+t}{1-t}\left[a_m\frac{c_i^s - c_m^s}{c_a^s} + h\frac{c_m^s}{c_a^s}\right],\tag{B12}$$

585

595

600

590

605 where  $\overline{a_{c\bar{s}}}$  (%) is the weighted discrimination for diffusion across the leaf boundary layer and inside the mesophyll, calculated as:

$$\overline{a_{c_i^s}} = \frac{a_b(c_a^s - c_s^s) + a_s(c_s^s - c_i^s)}{c_a^s - c_i^s},$$
(B13)

with  $c_s^s$ , the [COS] (pmol COS mol wet air<sup>-1</sup>) at the leaf surface, is:

$$c_s^s = c_a^s - A^s \frac{1.56}{g_h^w}. (B14)$$

610

The t is a ternary correction factor calculated as (Farquhar & Cernusak, 2012):

$$t = \alpha_{ac} \frac{E}{2g_t^s},\tag{B15}$$

where  $\alpha_{ac} = 1 + \frac{\overline{\alpha_{c_i^s}}}{1000}$ .

630

The  $a_b$  (= 3.5%),  $a_s$  (= 5.2%), and  $a_m$  (= 0.5%) are fractionations for COS diffusion across the boundary layer, 615 across the stomata, and due to COS dissolution and diffusion in water through the mesophyll, respectively (Davidson et al., 2022). h (=15 ± 2‰) is the fractionation during COS hydrolysis by CA (Davidson et al., 2022).

The  $c_m^s$  can be solved from Eqn 13 as:

$$c_m^s = \frac{(1-t) \cdot \Delta^{34} S \cdot c_a^s - \overline{a_{c_i^s}} (c_a^s - c_i^s) - (1+t) \cdot a_m \cdot c_i^s}{(1+t)(h-a_m)}.$$
 (B16)

Because  $t \cong 0$ , then Eq. (B16) can be simplified to:

 $c_m^s \cong \frac{\Delta^{34} \mathbf{S} \cdot c_a^s - \overline{a_{c_i^s}}(c_a^s - c_i^s) - a_m \cdot c_i^s}{h - a_m}.$ 620 (B17)

Substituting in Eq. (B17) the  $\overline{a_{c_i^s}}$  for its expression given in Eq. (B14) and rearranging terms result in:

$$c_m^s \cong \frac{c_a^s(\Delta^{34}S - a_b) + c_s^s(a_b - a_s) + c_i^s(a_s - a_m)}{h - a_m}$$
(B18)

Substituting in Eq. (B18) the fractionation factors by their values results in:

$$c_m^s \cong \frac{(\Delta^{34} S - 3.5) c_a^s - 1.7 c_s^s + 4.7 c_i^s}{14.5}, \tag{B19}$$
 where  $\Delta^{34} S$  (%0) can be experimentally determined during measurements of gas exchange as (Evans *et al.*, 1986):

625

$$\Delta^{34}S = \frac{c_e^s}{c_e^s - c_a^s} \frac{\delta_a^{34} - \delta_e^{34}}{1 + \delta_a^{34} - \frac{c_e^s}{c_e^s - c_a^s} (\delta_a^{34} - \delta_e^{34})},$$
(B20)

where  $c_e^s$  and  $c_a^s$  are the mole of COS in mole of dry air in the air entering and going out the chamber, and  $\delta_e^{34}$  and  $\delta_a^{34}$  (per mil) are the  $\delta^{34}$ S isotope composition of the air entering and leaving the chamber, respectively. The term  $\frac{c_e^s}{c_e^s-c_a^s}$  is often represented as  $\zeta$ . The  $\delta^{34}$ S values in the numerator should be divided by 1000 (for example if  $\delta_a^{34} = 10\%_0$ , then 0.0010 should be used).

We present  $c_m^s$  values calculated including ternary (Eq. (B16)). Ignoring ternary overestimated  $c_m^s \sim 1\%$  at PAR = 200 and  $\sim 5\%$  at PAR = 600.

#### 635 Appendix C: Overview of CO34S plant isotope discrimination data

| Publication     | Plant species             | [COS] (nn           | nol | [CO <sub>2</sub> ]  | (µmol | PAR (µmol m <sup>-2</sup> | $^{34}\Delta$ |   | LRU  |   |
|-----------------|---------------------------|---------------------|-----|---------------------|-------|---------------------------|---------------|---|------|---|
|                 |                           | mol <sup>-1</sup> ) |     | mol <sup>-1</sup> ) |       | s <sup>-1</sup> )         | (‰)           |   |      |   |
| Davidson et al. | Scindapsus                | $0.53 \pm 0.02$     |     | $500 \pm 80$        |       | 15.7                      | 1.6           | ± | 0.7  | ± |
| (2022)          | aureus (C <sub>3</sub> )  |                     |     |                     |       |                           | 0.1           |   | 0.1  |   |
| Davidson et al. | Zea mayz                  | $0.53 \pm 0.02$     |     | $500 \pm 80$        |       | 15.7                      | 5.4           | ± | 1.7  | ± |
| (2022)          |                           |                     |     |                     |       |                           | 0.5           |   | 0.3  |   |
| Davidson et al. | Scindapsus                | $3.4 \pm 0.1$       |     | $2900 \pm 90$       | 0     | 15.7                      | 4.9           | ± | 2.4  | ± |
| (2022)          | aureus (C <sub>3</sub> )  |                     |     |                     |       |                           | 0.5           |   | 0.3  |   |
| Davidson et al. | Zea mayz (C4)             | $3.4 \pm 0.1$       |     | $2900 \pm 90$       | 0     | 15.7                      | 9.2           | ± | 1.0  | ± |
| (2022)          |                           |                     |     |                     |       |                           | 0.4           |   | 0.1  |   |
| Baartman et al  | Helianthus                | $2.2 \pm 0.02$      |     | 434 ± 1             |       | 200                       | 3.6           | ± | 5.2  | ± |
| (this study)    | annuus (C <sub>3</sub> )  |                     |     |                     |       |                           | 1.2           |   | 0.16 |   |
| Baartman et al  | Helianthus                | $2.2 \pm 0.02$      |     | 434 ± 1             |       | 400                       | 3.7           | ± | 3.1  | ± |
| (this study)    | annuus (C <sub>3</sub> )  |                     |     |                     |       |                           | 0.4*          |   | 0.11 |   |
| Baartman et al  | Helianthus                | $2.2 \pm 0.02$      |     | 434 ± 1             |       | 600                       | 2.8           | ± | 2.3  | ± |
| (this study)    | annuus (C3)               |                     |     |                     |       |                           | 0.6           |   | 0.08 |   |
| Baartman et al  | Helianthus                | $2.2 \pm 0.02$      |     | 434 ± 1             |       | 0                         | 4.7           | ± | -    |   |
| (this study)    | annuus (C3)               |                     |     |                     |       |                           | 0.4*          |   |      |   |
| Baartman et al  | Helianthus                | $2.2 \pm 0.02$      |     | 434 ± 1             |       | 0                         | 1.3           | ± | -    |   |
| (this study)    | annuus (C3)               |                     |     |                     |       |                           | 0.4*          |   |      |   |
| Baartman et al  | Cyperus                   | $2.4 \pm 0.04$      |     | $427 \pm 0.5$       | i     | 200                       | 2.5           | ± | 3.0  | ± |
| (this study)    | papyrus (C4)              |                     |     |                     |       |                           | 0.4*          |   | 0.11 |   |
| Baartman et al  | Cyperus                   | $2.4 \pm 0.04$      |     | $427 \pm 0.5$       |       | 400                       | 2.6           | ± | 1.7  | ± |
| (this study)    | papyrus (C <sub>4</sub> ) |                     |     |                     |       |                           | 0.4           |   | 0.06 |   |

<sup>\*</sup>n = 1, error states is the single measurement precision instead of the repeatability precision

# Data availability

640

645

The dataset is available at: 10.5281/zenodo.14677494

### **Author contribution**

Conceptualization: SLB, MCK, MEP, LW. Data curation: SLB. Formal analysis: SLB, NUL. Funding acquisition: MCK. Investigation: SLB, SMD, MW, LMJK, LM, AC, SH. Methodology: SLB, SMD, MW, LMJK, MEP. Resources: SMD, MW, LM, SH. Supervision: MEP, TR, MCK. Visualization: SLB, NUL. Writing – original draft preparation: SLB, NUL. Writing – review & editing: SMD, MW, LMJK, NUL, LM, MEP, AC, LW, TR, SH, MCK.

# **Competing interests**

The authors declare that they have no conflict of interest

# 650 Acknowledgements

We are grateful for the technical support from Carina van der Veen, Marcel Portanger and Giorgio Cover. The authors gratefully acknowledge the insightful discussions with Jérôme Ogée.

# Financial support

This project has received funding from the European Research Council (ERC) under the European Union's Horizon 2020 research and innovation program under grant agreement No 742798 (COS-OCS; to M. C. Krol)

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
