# Peer review of "Isotope discrimination of carbonyl sulfide (34S) and carbon dioxide (13C, 18O) during plant uptake in flow-through chamber experiments"

_EGUsphere, 2025_

## Author Response (AR1)

Author's response: Baartman et al., Isotope discrimination of carbonyl sulfide (34S) and carbon dioxide (13C, 18O) during plant uptake in flow-through chamber experiments

**Response to editor comments:**

Line 73 (previous version manuscript): "biofosfaat" changed to "biophosphate"

Table 1: added the international isotope reference scale for each reported isotope ratio

Table 2 (in previous version manuscript): removed this table in section 2.4. Table 3 from the previous version of the manuscript is now Table 2.

Equation (5): corrected the typographical mistakes and for all following equations (including the equations in appendix B), we changed the notations of entering and ambient air to be consistent. Entering air now always has a subscript "e" and the ambient air in the chamber has the subscript "a".

All equations and in text: we changed all occurrences of "C" to be capitalized, for consistency

Table 3 (previous version manuscript)/Table 2 (new version manuscript): Added "apparent" to  $^{18}\Delta$ . Switched the units for As and Ac. We also added new data to this table: number of samples, stomatal conductance, total conductance to COS, total conductance to CO2 and  $C_i^{\text{C}}/C_a^{\text{C}}$ , according to suggestions from the reviewers. We included errors on the COS and CO2 fluxes and the LRU. We also updated the Table caption, according to the suggestions from the reviewers.

Equation (B2): removed from Appendix B and subsequent equation numbering was updated. We instead refer to Equation (5).

In order to be more transparent and consistent about the uncertainties, we have changed the manuscript to have all the reported uncertainties on the averages represent the standard error of the mean (SEM), taking into account the errors on the individual measurements. And when we report the uncertainty on single measurements, we have indicated this.

We have implemented all the revisions to the manuscript that we committed to in our responses to the referee comments below.

**Response to comments by Referee #1**

We thank the reviewer for taking the time to read our manuscript and for providing useful suggestions for improvement. The reviewer comments and questions are in black and our responses are in blue below each comment.

The paper deals with an important and exciting topic. The use of COS as a unique tracer of photosynthesis and the rare measurements of the isotopic discrimination, D34S, associated with COS uptake. The paper presents a unique measurement system for gas exchange, COS, and isotopic analysis, and it is well-written.

However, the paper has some rather significant issues that need attention. This is partly so as there seems to be a gap between the impressive analytical measurements and the experimental, plant gas exchange, part . Below are some of the concerns noted as I was reading the paper (i.e., in no special order) that I hope will help to improve the paper.

In general, the motivation is to introduce D34S to "provide useful information on the COS uptake process and help to constrain the COS budget" (upfront in the abstract). However, at the end, the paper does not tell us what we learned in either aspect. **At least some discussion of these aspects is needed, or these should be strongly toned down.**

In fact, the paper goes on to declare another much more modest and specific goal: **To verify the published D34S data obtained in a 'closed system' (Davisson et al.) in their new 'open steady-state system'**. The paper generally confirms the earlier data but in a way that does not provide additional confidence due to the experimental difficulties. In its present form, therefore, it is uncertain whether the paper will advance the field in that respect. **Better focusing on what exactly is the bottom line/take-home message for D34S is needed.**

In the revised manuscript, we will clarify the goal of the manuscript with the reviewer's helpful comments. We agree that measuring plant COS discrimination poses technological and interpretational challenges, but we believe it holds potential to inform us about processes occurring across scales, from the leaf (e.g., test if COS is a unidirectional flux into the leaf) to the globe (e.g., to partition COS oceanic and anthropogenic sources using isotope constrained COS tropospheric mass balances (Davidson *et al.*, 2021)). Additionally, more observational data are required to improve and test mechanistic models of plant COS discrimination. The objective of this work was to design a system to simultaneously measure COS and CO2 gas exchange and isotope discrimination using a continuous-flow plant chamber. We describe this system, present a first set of measurements in C3 and C4 plants, and compare them with the only other available dataset obtained with a closed plant chamber (Davidson *et al.*, 2021; Davidson *et al.*, 2022).

To improve clarity and balance, we will revise the manuscript to better reflect study's objectives and the scope of the results. We will include a perspectives section (see suggestions from Referee #2) with ideas for future research. In a subsequent paper that we are currently writing, we use all available measurements to develop a mechanistic model for COS plant discrimination. In the revised manuscript, we will use some of the information from this accompanying paper to provide context for the measurement results.

On the methodological side, it is not clear how many plants were used as replicates. In the Method, three papyrus cuttings and "a sunflower plant" were noted. In Fig. 3, n=2 is indicated (with SE...). In Fig. 4, no replication is indicated; in Fig. 5, 6, some individual replications are plotted, each with its own SE. While the information is incomplete and confusing, the impression is that only a few actual replications were made, and there is no clear distinction between the precision (repeating the measurements) and replications. There is also missing information on Blank Testing of the chambers, which seems to be critical in COS experiments. The inlet COS concentration (2-3 ppb) is 4-5 times the atmospheric level) is indicated but not the chamber ambient concentrations (outlet). [BTW, in the Abstract, fluxes are reported in pmo mol-1, which are not flux units.] **More information seems to be required.**

We will include the requested information in the revised document. The data are from two experiments; one with a single sunflower and one with an assemblage of papyrus leaves. At each light level, duplicate samples were collected within the same experiment. We report their averages and standard errors. A few samples were likely affected by contamination and excluded from the analysis.

For the papyrus under dark conditions (PAR=0), the COS uptake was too low to calculate a discrimination value. As a result, these data points are not shown in the discrimination plots but are included in the flux figures. We will clarify the meaning of the duplicates, missing values and the errors in the Methods section and in the caption of Table 3 and relevant

figures. Table 3 will also be expanded to include chamber mole fractions and stomatal conductance for each plant and treatment, as suggested in a later comment.

Blank (empty chamber) tests were conducted prior to the plant measurements, though not previously mentioned. We will add a description and the results of these tests in the revised manuscript.

The flux units in the Abstract should have been pmol m-2 s-1, this was a typographical error, which we will correct.

The aspects noted above are significant as many of the observations are somewhat unexpected or uncharacteristic, and a range of particular explanations are required, such as non-uniform light level ("low light" in parts), "not optimal behavior"; "stomata not fully open"; increasing Ci with increasing light and increasing A, no response of COS assimilation to light, mostly constant D34S, constant Cm, etc. In fact, the feeling is that more measurements would help to get more conventional results.

We agree that more measurements would have strengthened our study, but logistic constraints limited extending the measurements. Furthermore, the isotope system's capacity further constrained the amount of samples that we could measure (given that it takes a full day to measure only 3 - 4 samples).

We did conduct a follow-up experiment with the goal of expanding the dataset. However, when the experiments were finished, the COS isotope system needed extensive maintenance, after which we were limited in time and personnel. In addition, storage issues may have had an influence on these samples as COS can be unstable during longer storage.

Despite these limitations, we believe that the current dataset and innovation in methodology provide valuable insights, given the scarcity COS isotope discrimination data. At present, additional measurements are not feasible. The COS isotope system in Utrecht is non-operational and a new system in Bordeaux is still under development.

Fig. 3 presents a nearly complete insensitivity of COS uptake to light level (in sharp contrast to CO2 uptake), and it is explained by the light in-sensitivity of carbonic anhydrase. However, COS should still respond to light for the same CA activity because of its effect on conductance (g). **No information on conductance is given in this paper**.

We will provide the values for stomatal conductance and total conductance in the revised manuscript. We appreciate the reviewer's point - information on these parameters is needed to explain the (lack of) variability with light level of the uptake fluxes.

Briefly, in non-dark conditions (PAR>0), stomatal conductance remained above 0.25 mol m-2 s-1, suggesting that stomatal opening was sufficient to maintain COS uptake, even at lower light levels. This is also supported by the relatively small changes observed in Cis and Ci/Cas, between the highest and the lowest light setting. A more detailed explanation will be included in the revised manuscript.

Details of leaf gas exchange equations are presented, including conductance, internal concentrations, etc. However, all those were developed strictly for the leaf scale, which may not apply here. The photo in the Appendix shows that this was a rather 'dense canopy scale' experiment. The authors note this can explain some of the non-typical observations, but there is no discussion on how to scale from leaf to canopy (or vice versa). The photo clearly indicates no uniformity in conditions and, in turn, in activities. **This scaling gap should be addressed, and if it can be overcome, it should be explained in more detail**. By the

way, it seems there are some publications on branch scale measurements, which can be helpful to compare (likely also Yang et al. 2017 or 2018 who tried to scale between leaf to canopy).

We recognize that our gas-exchange approach is relatively simple and involves certain assumptions. We applied a *big leaf* approach, which we consider the most appropriate for our setup, given the use of small plants and a well-ventilated chamber thereby - minimizing boundary layer effects.

We also recognize that likely not all leaves received the same amount of PAR because of shading. However, given the precision at which COS isotope discrimination can be currently determined, it would seem too complex to go beyond a *big leaf* approach in our study. In addition, we did not obtain the gas-exchange data for the stem of the plants, so these could not be included in our calculations.

We will add a section to the Methods in the revised manuscript explaining our choice of the *big leaf* approach and the assumptions involved.

Along these lines, internal concentrations (Ci) are estimated using the leaf-scale equations, and Cm is calculated based on the D34S estimates. **The difference between the Ci and Cm is interesting but not defined or discussed**, except that very different values are reported for Ci/Ca and Cm/Ca.

We calculated the Cm/Ca ratios for COS as the mesophyll cell is the end-point for COS assimilation (the location of CA). We were also interested in whether Cm/Ca values could help explain the limited variation in COS flux across light levels, as. Indeed, we observed little variability in Cm/Ca with changing light. We will expand the discussion on the differences between Ci/Ca and Cm/Ca in the revised manuscript.

Note also that the physiological calculations of conductance, g, based on E and ci, depend on leaf temperature and water vapor saturation assumption. This is tricky in the present study, which uses a dense canopy in a different light and temperature in the chamber. It seems that COS flux, as long as it is based on the assumption of near-zero internal concentrations (i.e., **no compensation point, an issue that is ignored in this paper**), may offer a simpler alternative to total conductance, which could at least be compared (i.e., As=gCa...).

We appreciate the reviewer's thoughtful concerns. The equations by von Caemmerer and Farquhar (1981) do assume saturation of the leaf internal airspaces with water vapor - an assumption that may not hold under high evaporative demands (Cernusak *et al.*, 2018; Cernusak *et al.*, 2024), though such conditions were not present during our measurements.

Our calculations of conductance to COS represent a canopy average and may carry uncertainties related to leaf temperature. However, estimating the conductance under the assumption of zero COS concentration in the mesophyll could introduce even greater uncertainty

As mentioned earlier, we are preparing a companion paper that presents a modelling framework for COS isotope discrimination in plants. This model can account for non-zero internal COS concentrations and emissions, and will allow us to explore their effects on observed isotope discrimination.

Cernusak, L. A., Ubierna, N., Jenkins, M. W., Garrity, S. R., Rahn, T., Powers, H. H., ... & Farquhar, G. D. (2018). Unsaturation of vapour pressure inside leaves of two conifer species. *Scientific reports*, 8(1), 1-7.

Cernusak, L. A., Wong, S. C., Stuart-Williams, H., Márquez, D. A., Pontarin, N., & Farquhar, G. D. (2024). Unsaturation in the air spaces of leaves and its implications. *Plant, Cell & Environment*, 47(10), 3685-3698.

The LRU estimates are important. However, it is clearly sensitive to the high ambient COS used as COS uptake can generally be linearly related to Ca-cos, and it also seems that some information on this response may be available in the literature. In this case, **the effect could be estimated to some extent, and an attempt to correct the LRU for comparison with literature values at ambient COS could be made,** and discussed. In fact, a good agreement on the uncorrected LRU does not add confidence, as noted above.

Although the primary goal of our experiments was not to quantify LRU under natural conditions, we appreciate this suggestion. Applying such a correction would rely on a limited dataset (Stimler et al. 2011) and could introduce additional assumptions and uncertainties. Nonetheless, we will acknowledge in the revised manuscript that the LRU values reported may not fully represent natural conditions due to the use of higher-than-ambient COS mole fractions in the chamber air.

**Response to comments by Referee #2**

We thank the reviewer for reviewing our manuscript and for providing useful suggestions for improvement. The reviewer comments are in black and our responses are in blue below each comment.

**General**

The paper presents the first isotopic measurements of COS and CO2 made in flow-through chamber for one papyrus, a C4 plant and sunflower, a C3 plant. The setting allows them to derive the COS and CO2 plant uptake, the internal and mesophyll concentrations of CO2 and COS respectively, the photosynthetic discrimination against 13CO2, C18O2 and to explore how they respond to increasing light intensity. They conclude about a distinct behavior was observed between the C3 and the C4 plant.

The strength of the paper stands on the perspectives that such isotopic measurements could provide insights into the underlying processes of the COS and CO2 plant uptake. Especially, the isotopic discrimination of COS gives insight into the CA activity. However, the paper needs to be written in a rush, and the authors should add more contexts and perspectives. In short, I would recommend publishing this paper with major revisions, which consist in improving the storyline. The comments below go around those lines.

The storyline (objectives, method, conclusion) needs to be clarified: . Which complementary piece of information each isotope discrimination can bring? The benefits of using isotopes discrimination of carbon and COS are not clearly explained. Also, in the method section, the photosynthetic discriminations against 13CO2 and C18O2 are not presented. The author should also some perspectives in their conclusion.

In the revised manuscript, we will clarify the overall storyline and more explicitly define the objectives and benefits of combining  $CO_2$  and COS isotope discrimination measurements. Additionally, we will provide more detail on discrimination against  $^{13}CO_2$  and  $C^{18}O^{16}O$ . However, we intend to keep the primary focus on COS isotope discrimination and will refer to relevant literature for in-depth discussions on  $CO_2$  isotope discrimination in plants.

Clarity of the Figures: The green and the dark green colors are hard to distinguish. I would recommend putting the same colors for each PAR for papyrus and sunflower. The number of plant replicate is confusing. What is a replicate? I would suggest making a tabular with the

number of measurement/replicates for each measured or computed quantity. Likewise, the observations made at PAR equal to 0 are not shown on Figure 4.

We thank the reviewer for the suggestions. We will include the information on the number of measurements/replicates for each measured plant and treatment.

Regarding the replicates, these were air samples for isotope analysis taken consecutively during the same treatment (PAR level) for the same plant. We will clarify this in the revised manuscript.

For the figures, we would still like to maintain the distinction between the different PAR levels for the two plant species, as this provides valuable information that would otherwise be missed. In the revised manuscript, we will improve the color scheme for clarity.

The data for PAR = 0 are not included in Figure 4 because the figure plots the CO2 uptake against Cis/Cas, and in the dark, the plant was not taking up CO2 uptake, but respiring it. We will address the absence of this data in the caption of Figure 4 in the revised manuscript.

Representativeness of the experiments: The papyrus, a C4 plant, in the experiment grows in tropical swamps and in arid light saturated environment. How this C4 plant is representative of all the C4 plants? The experiment is far from reality as, because of time constrain, the author did not repeat the experiments with higher light intensity than PAR=400. The effects of soil water level, VPD and nutrients availabilities are also not discussed. Likewise, the chamber was well aerated, which results in infinite boundary layer conductance. Is it realistic, especially at night when the boundary layer becomes stable?

We acknowledge that papyrus is not a widely studied C4 species and may not be fully representative of typical C4 plants. Given the constraints we faced, we used what was available and feasible. We did attempt to grow and measure several other C4 species including *Zea mays* (a widely studied C4 species), but encountered significant experimental challenges. For example, successfully quantifying COS isotope discrimination in our flow-through gas-exchange system, required significant COS uptake while maintaining an adequate flow rate to minimize boundary layer and condensation issues. We therefore needed relatively large leaf areas, which we were not able to obtain from other C4 species.

We agree that future experiments would benefit from using representative C4 species and we will include this suggestion in the outlook section of the revised manuscript.

Due to the time-consuming nature of the COS isotope measurements (a maximum of 3 samples per day under optimal conditions), we limited the number of samples to 18.

Regarding the dark measurements and the infinite boundary layer conductance; we were not aiming to perfectly replicate night conditions, but rather to observe the effect of stomatal closure on COS and CO2 fluxes and isotope discrimination.

More in-depth analysis of the results is needed: Stomatal and total conductances are needed to explain the results for the C3 and C4 plants (as done in Stimler et al. 2011). The paper only presents experimental results and lacks interpretation in terms of processes.

We agree with the reviewer that a more detailed analysis, including stomatal conductance, would strengthen the paper. In the revised manuscript, we will expand the Results & Discussion to incorporate the conductance data and provide a more detailed interpretation of the isotope discrimination results, as also mentioned in our response to Referee #1.

Specific comments

**Abstract**

Line 24 – "does not exit the leaf again". Confusing, sounds like it is the same CO2 molecule that enters and leaves the leaf.

**We will rephrase this**

Line 25 – Precise why performing such isotopic measurements and how can isotopic discrimination of COS **and CO2**. Only the isotopic discrimination of COS is mentioned.

We will include a sentence to mention the motivation behind measuring both COS and CO2 fluxes and isotope discrimination.

Line 30 -35: Mention what are the implications of these results

We will add a sentence on the implication of the results

Line 35: "The papyrus was not ... experiments" Explain a bit better. How is the C4 plant supposed to behave? And how does the papyrus behave in the experiment?

We will add an explanation of what would be expected from a typical C4 plant in terms of CO2 fluxes and isotope discrimination.

**Introduction**

• You only describe the processes and the equations underlying the isotope discrimination of COS. As the title mentions it, you should also describe, here or in the method, the processes underlying the isotope discrimination of CO2.

As the primary focus of this paper is COS isotope discrimination, a relatively newly studied topic, we chose to introduce the relevant calculations and definitions in the introduction. CO2 discrimination values are to complement and contextualize the COS data. Given that CO2 isotope discrimination is a well-established field with widely understood methodologies, we decided not to go into detail on its method.

We agree, however, that the mention of  $CO_2$  isotope measurements in the final paragraph of the introduction (lines 110-114), comes slightly out of the blue. In the revised manuscript, we will clarify the benefit of including  $CO_2$  isotope data alongside the COS measurements in the introduction.

 You should add a tabular or a section showing which piece of information each isotope discrimination or molecule can give to guide the reader.

We appreciate the suggestion of the reviewer to clarify what information can be gained from each isotope discrimination value (COS and CO2). However, we believe a table may not be the most effective way to present this, as the interpretation of each discrimination value varies between C3 and C4 species. Additionally, COS isotope discrimination research is still in its early stages, and it may be difficult to summarize in a table exactly what information each isotopologue could provide. For instance, variations in observed sulfur-34 discrimination in COS result from a combination of factors, including conductances and CA activity.

In the revised manuscript, we will include a section in the introduction highlighting the potential insights that can be gained from each isotope discrimination value.

• Make a tabular with the Davidson et al. (2022) to compare their measurements with yours, add a column for the experimental conditions (CO2, COS mole fractions ec)

We appreciate this suggestion and we will include a table in the Appendix of the revised manuscript. While Davidson et al. (2021; 2022) used different chamber conditions - such as  $CO_2$  and COS mole fractions, as well as different temperatures, and significantly lower PAR - which may limit direct comparisons, we agree that providing an overview of all currently available data can be valuable for the reader.

Line 40 Add a citation

**We will do this**

Line 43 The SIF, satellite retrievals of GPP, net co2 fluxes estimated by atmospheric inversion are not mentioned. Which advantage has COS plant uptake compared to these mentioned methods?

We appreciate this, however we specifically chose to limit to and name methods based on gas exchange and flux. We indeed need to study GPP from various perspectives. Our point is not so much that COS is more advantageous, but that COS may be a potential valuable independent proxy that can further constrain existing estimations of GPP. Promising approaches to estimate GPP on the regional and global scale include SIF, NirV, and atmospheric inversions e.g. using COS (Ma et al., 2021; Remaud et al. 2022). However, the latter method requires the use of prior information. Therefore, studies on ecosystem and plant scales are needed to obtain a better mechanistic understanding of the combined exchange of water, CO2, and COS.

Ma, J., Kooijmans, L. M., Cho, A., Montzka, S. A., Glatthor, N., Worden, J. R., ... & Krol, M. C. (2021). Inverse modelling of carbonyl sulfide: implementation, evaluation and implications for the global budget. *Atmospheric Chemistry and Physics*, *21*(5), 3507-3529.

Remaud, M., Chevallier, F., Maignan, F., Belviso, S., Berchet, A., Parouffe, A., ... & Peylin, P. (2022). Plant gross primary production, plant respiration and carbonyl sulfide emissions over the globe inferred by atmospheric inverse modelling. *Atmospheric Chemistry and Physics*, 22(4), 2525-2552.

**Line 45 Add Wehr et al. 2015**

(https://www.sciencedirect.com/science/article/abs/pii/S0168192315007145) who quantified respiration from GPP thanks to isotopic measurements.

We will do this

Line 49 Lack of fundamental papers about COS

We will add these

Line 52 Cite Montzka et al. 2007 https://agupubs.onlinelibrary.wiley.com/doi/full/10.1029/2006JD007665

**We will do this**

Line 56 «Assuming that there is no COS emissions" Discuss the validity of this assumption: it has been shown that Beliso et al.

(2018) (https://journals.plos.org/plosone/article?id=10.1371/journal.pone.0278584) showed that rapeseed emit COS. Some plants in swamps also emit COS.

Indeed, recent evidence suggests that certain species and/or environmental conditions may lead to COS emissions or the existence of a COS compensation point. We will add a brief discussion of this in the revised manuscript.

Line 62: How negligible is the daytime respiration? (https://www.sciencedirect.com/science/article/abs/pii/S0168192315007145)

Daytime respiration is indeed not always negligible. One could also account for daytime respiration when using LRU as a tool for estimating GPP. We will adjust this sentence accordingly. However, a detailed discussion of the broader caveats associated with using LRU for GPP estimation falls outside the scope of this manuscript.

Figure 70 Please make the Figure more understandable. What is the blue and red lines in the middle? What are the zigzag lines in the middle? Add the name of the conductance's. Some accolades near the name of the space could help to separate the cell/space.

We will revise this figure to make it more understandable and include labels for the conductances and other parameters. We will also include a more thorough explanation in the figure caption.

Line 75 The first two sentences, "COS discrimination...factors" and "The discrimination ..." are not logical. You should explain which beneficial information COS isotope discrimination can give based on literature and go to the next line, for the definition.

We agree that this section would benefit from a clearer explanation of the value of COS isotope discrimination data. We will revise the manuscript to include this, as also noted in our responses to earlier comments.

Line 88 Add As the reaction with CA is supposed to be irreversible,

We will add this

Line 93 Explain why this may be a too crude simplification of the diffusion processes taking place based on literature studies (<a href="https://bg.copernicus.org/articles/20/2573/2023/bg-20-2573-2023.pdf">https://bg.copernicus.org/articles/20/2573/2023/bg-20-2573-2023.pdf</a>)

We will add this

Line 95 In this case = for C4 species?

We intended to refer to the case where mesophyll COS concentration is not zero. We will rephrase this in the revised manuscript.

Line 101 "ambient COS and CO2 concentrations". You just said above the experiments were caried at high CO2 and COS mole fractions. And how high?

Davidson et al. (2021; 2022) conducted several experiments using both elevated and ambient COS and CO2 concentrations. We will include the relevant concentration values in the revised manuscript. The elevated concentration measurements were specifically designed to estimate the carbonic anhydrase (CA) isotope fractionation.

Line 165 Why do you use to instruments to measure both air in and out of the chamber? What is the specifities of each instrument?

We measured both inlet and outlet air to calculate the fluxes. The LI-COR instrument was part of the existing chamber setup and was used specifically to measure the  $H_2O$  fluxes. Since the LI-COR cannot detect COS, we used a second instrument, the QCLS, which measures COS mole fractions with high precision. We will make this more explicit in the improved manuscript.

Line 182 Why choosing these temperatures? Are they representative of the place where these two plants live or is it associated with maximal CA activity?

We selected relatively high temperatures to obtain sufficient COS uptake fluxes for isotope discrimination analysis, while also avoiding condensation in the system. Throughout the experiments, we maintained the temperature as consistently as possible.

Line 185 remove the space

We will do this

Line 189 add coma

We will do this

Line 194 There is a mistake in the formula, please verify. Wa is not wo?

There is indeed a typographical mistake here. We will make sure to fix it

Line 261 – 264 These two sentences should be in the method. This is not clear either. Why some sample must be treated as duplicate?

We will move this section to the Methods. At each treatment, we collected two air samples for isotope analysis. Under light conditions (PAR > 0), fluxes remained stable, so these samples were treated as duplicates and averaged. However, under dark conditions (PAR = 0), the plant was still adjusting, and fluxes were gradually changing. As a result, the two samples captured different states and were treated as separate data points rather than duplicates. We will clarify this approach more explicitly in the revised manuscript.

Line 264 Can you compare your COS uptake fluxes with those from the literature by making a distinction between C3/C4 plant? Can you find some COS uptake flux measured at the ecosystem level (fluxnet type)?

We thank the reviewer for the suggestion. We will include a brief literature comparison section distinguishing between  $C_3$  and  $C_4$  COS fluxes. However, we note that there is limited COS flux data available for  $C_4$  species, particularly at the ecosystem scale.

Line 266 Some measurements are also made at the ecosystem level and not in controlled chamber. Please mention that.

The sentence mentions that some studies are at the ecosystem scale, but it indeed does not distinguish between studies at the canopy scale and those at the ecosystem scale. We will clarify this distinction in the revised manuscript.

Figure 3. Why the As is lower in C4 plant than in C3 plant at PAR=0?

The difference can be attributed to lower stomatal conductance for C4 at the time of sampling compared to the C3 species. Since we plan to include conductance data in the revised manuscript, we will also provide an explanation of this phenomenon in the paper.

Line 289 Would be nice to have a plot for LRU as well on Figure 3 (3 panels)

We appreciate this suggestion and will include this in the revised manuscript.

Line 294 Please make a distinction between C3 (LRU=1.68) and C4 plant (1.21). only 4 values for C4....

We thank the reviewer for pointing this out and we will make the distinction in the revised manuscript.

Line 297 Your results cannot be directly compared with Davidson 2022 as they used both high CO2 and high COS concentrations. For instance, Wu Sun 2021 showed that LRU increases with CO2 and that can also affect Davidson 2022 results.

We appreciate this suggestion and will mention this in the revised manuscript.

Line 302 Cite Wu Sun for the dependance of LRU to various environmental conditions and be more specific.

We will do this

Line 304 Explain first why the quantity ci/ca is interesting to study.

For CO2, Ci/Ca is indeed not explicitly explained. We will include a brief explanation of the relevance of studying Ci/Ca for CO2 in the revised manuscript.

Line 370 What is the typical PAR that other experiments use?

We are not entirely clear on the reviewer's question. Low PAR values are mentioned in the comparison with studies in line 368. Many studies involving plant chamber experiments typically perform light-response curves starting from around PAR = 100 (or similar) up to PAR = 1500 or 2000  $\mu$ mol m-2 s-1.

Line 371-377, not clear, explain better...As papyrus grow in swamps, this setting is closer to reality?

We acknowledge that the conditions under which we measured the papyrus leaves were less than ideal. We initially attempted to measure several other  $C_4$  species, but the COS uptake was too low for precise isotope discrimination. Given the circumstances, we worked with what was available - papyrus from the tropical greenhouse at Wageningen University. It should be noted that these plants were grown in a relatively dry and warm environment, not a swamp. Due to the plant's size, we cut several stems with leaves and placed the stems in water to transport to the lab. Papyrus clearly thrived when kept in water and continued photosynthesis shortly after being placed in the chamber. However, we observed some unexpected  $CO_2$  flux and isotope discrimination values that were atypical for a  $C_4$  species. We hypothesize that some  $CO_2$  may have been transported from the water into the chamber, and we will clarify this in the revised manuscript.

Line 435 Remind that CO2 assimilations in C4 plants are expected to be more efficient

We are not sure which sentence the reviewer is referring to, as line 435 is the caption of Figure A1 in the appendices. If the reviewer is referring to line 415, we can clarify this paragraph by specifically discussing the differences in  $CO_2$  assimilation between our  $C_3$  and  $C_4$  species.

Line 421 These two sentences are contradictory. First, you sat that the CA reaction is light independent and then you say that the COS uptake was lower during the dark experiment.

We will rephrase this sentence to clarify that COS assimilation by carbonic anhydrase (CA) is light-independent. However, stomatal opening is still necessary for COS uptake, so when stomata partially close in the dark, COS uptake will decrease.

**Conclusion**

Lower discrimination against C18O2 is observed, which is consistent with previous measurements (Stimler et al., 2011). Why not for 14COS?

We are not entirely sure what the reviewer means by 14COS, but we assume they are referring to  $CO^{34}S$ . Additionally, we believe the reviewer is referring to our observation of lower  $C^{18}O^{16}O$  in  $C_4$  compared to  $C_3$  species, which aligns with the findings of Stimler et al. (2011). The question seems to be why we did not observe the same pattern for  $CO^{34}S$  discrimination. We address these unexpected results in lines 424-427 in the conclusion. We also observed atypical results for  $^{13}CO_2$  discrimination in our  $C_4$  papyrus, highlighting the value of measuring both COS and  $CO_2$  isotope discrimination in a single experiment.

Can you bring some conclusions about the CA activity?

We did not directly measure CA isotope discrimination, and the cm/ca values we report are derived from the sulfur isotope discrimination, meaning they are not independent. Therefore, with our current dataset, we can only speculate on CA activity. To better constrain this in future studies, targeted measurements of CO34S isotope discrimination by the enzyme CA would be highly valuable. We will add this recommendation to the perspectives section of the revised manuscript.

It is necessary to add a perspective section to pave the way for future experiments.

We appreciate the suggestion and we will make sure to add this to the revised manuscript.

---

## Editor Decision (ED1)

**Isotope discrimination of carbonyl sulfide (34S) and carbon dioxide (13C, 18O) during plant uptake in flow-through chamber experiments**

Sophie L. Baartman1,2, Steven M. Driever3, Maarten Wassenaar4, Linda M.J. Kooijmans2, Nerea

Ubierna Lopez6, Leon Mossink3, Maria E. Popa2, Ara Cho2, Lisa Wingate6, Thomas Röckmann1, Steven M.A.C. van Heuven5, Maarten C. Krol1,2

1Institute for Marine and Atmospheric Research Utrecht (IMAU), Princetonplein 5, 3584 CC Utrecht, The Netherlands

2Meteorology and Air Quality, Wageningen University and Research Centre, Droevendaalsesteeg 4, 6708 PB

Wageningen, The Netherlands

3Centre for Crop Systems Analysis, Wageningen University and Research Centre, Droevendaalsesteeg 4, 6708 PB Wageningen, The Netherlands

4Horticulture and Product Physiology, Wageningen University and Research Centre, Droevendaalsesteeg 4, 6708 PB Wageningen, The Netherlands

5 Centre for Isotope Research (CIO), Energy and Sustainability Research Institute Groningen, University of Groningen, 9747 AG Groningen, the Netherlands

6Institut National de la Recherche Agronomique, Bordeaux Sciences Agro, Unité Mixte de Recherche 1391, Interactions Sol-Plante-Atmosphère, 33140 Villenave d'Ornon, France

Correspondence to: Sophie L. Baartman (sophie.baartman@wur.nl)

20

25

30

35

**Abstract.** Carbonyl sulfide (COS) has been proposed as a proxy for gross primary production (GPP), as it is taken up by plants through a comparable pathway as CO2. COS diffuses into the leaf and undergoes an essentially one-way reaction in the mesophyll cells, catalyzed by the enzyme carbonic anhydrase (CA), and does not exit the leaf again. In order to use COS as a proxy for GPP, however, the mechanisms of COS uptake and its coupling to CO2 uptake need to be well understood. Characterizing the isotopic discrimination of COS during plant uptake can provide useful information on the COS uptake process and can help to constrain the COS budget.

This study presents joint measurements of isotope discrimination during plant uptake for COS (CO34S) and CO2 (13CO2 and C18O16O). A C3 plant, sunflower (*Helianthus annuus*), and a C4 plant, papyrus (*Cyperus papyrus*), were enclosed in a flow-through plant chamber and exposed to varying light levels. The incoming and outgoing gas compositions were measured online, and discrete air samples were taken for isotope analysis.

The COS uptake flux was around 75 pmol mol-1 for sunflower and between 99 and 110 pmol mol-1 for papyrus. The corresponding  $^{34}\Delta$  for COS was  $3.4\pm0.8$  % for sunflower and  $2.6\pm0.3$  % for papyrus. For CO2, a negative relationship was observed between the uptake flux and the isotopic discriminations  $^{13}\Delta$  and  $^{18}\Delta$ . The CO2 uptake and  $\Delta$  values indicate that our sunflower behaved as expected for a C3 plant, while the papyrus was not displaying typical C4 behavior, perhaps due to the relative low light conditions during our experiments.

45

50

55

60

**1. Introduction**

Photosynthetic uptake of CO2 by the terrestrial biosphere, quantified by the gross primary production (GPP), is the largest sink of atmospheric CO2, and may be altered as the climate changes. For making accurate future climate projections, it is important to quantify changes in the functioning of the biosphere and its influence on the atmospheric composition. Several techniques can be used to quantify photosynthesis and respiration fluxes at the ecosystem and larger scales, such as Eddy Covariance (EC) (Asaf et al., 2013; Billesbach et al., 2014; Commane et al., 2015; Wehr et al., 2017; Vesala et al., 2022) or variations in the stable isotopic composition of CO2 (e.g. Farquhar and Lloyd, 1993; Farquhar et al., 1993; Wingate et al., 2007; Gentsch et al., 2014). However, these techniques have limitations, because they either measure net CO2 fluxes (Wohlfahrt et al., 2012; Kooijmans et al., 2017) or they require additional measurements such as the oxygen isotope composition of water pools (Wingate et al., 2010; Adnew et al., 2020). Because of these limitations, other potential independent proxies for GPP have recently gained attention, especially the trace gas carbonyl sulfide (COS or OCS, COS henceforth) (Whelan et al., 2018; Lai et al., 2024).

COS is the most abundant sulfur-containing atmospheric trace gas, with a tropospheric mole fraction of around 500 ppt that displays a strong seasonal cycle, mostly due to the uptake of COS by terrestrial vegetation during photosynthesis. Figure 1 shows a schematic of the uptake pathways and assimilation locations of COS and CO2 in the leaf. Similarly to CO2, COS diffuses across the leaf boundary layer, through the stomata and into the leaf mesophyll cells (Protoschill-Krebs and Kesselmeier, 1992; Protoschill-Krebs et al., 1996). There, COS is hydrolyzed in an essentially one-way reaction, catalyzed by the enzyme carbonic anhydrase (CA), in contrast to the reversible hydration reaction that CO2 undergoes (Protoschill-Krebs and Kesselmeier, 1992; Protoschill-Krebs et al., 1996). Assuming that there is no COS emission, the COS uptake by plants is proportional to photosynthetic uptake of CO2, and therefore, GPP can be derived from the leaf-scale relative uptake ratio (LRU) of COS and CO2 uptake fluxes,  $A^S$  (pmol mol-1) and  $C_a^C$  (µmol mol-1) using Eq. (1):

$$LRU = \frac{A^S}{A^C} * \frac{C_a^C}{C_a^S} \tag{1}$$

If we assume negligible daytime leaf respiration,  $A^{C}$  can be replaced by GPP, which can then be estimated using Eq. (2) (re-arrangement of Eq. (1)).

$$GPP = A^{S} \frac{C_a^{C}}{C_a^{S}} * \frac{1}{LRU}$$
 (2)

While the use of LRU as a link between COS and CO2 fluxes seems promising, some studies have shown that the LRU is not constant among species and changes with environmental conditions such as photosynthetically active radiation (PAR), temperature and vapor pressure deficit (VPD) (Kooijmans et al., 2019; Maignan et al., 2021; Spielmann et al., 2023; Sun et al., 2024). Thus, a more thorough understanding of the physiological drivers and limitations of COS uptake by plants, and its relationship with CO2 uptake, is needed.

75

80

85

90

95

Figure 1. Schematic (simplified) representation of the diffusion pathways of  $CO_2$  (left) and COS (right) into a  $C_3$  leaf, including the mole fractions of both species in the atmosphere ( $C_a$ ), the intercellular space ( $C_i$ ), the mesophyll cell ( $C_m$ ) and, for  $CO_2$ , the chloroplast ( $C_c$ ). The enzymes ribulose-1,5-bifosfaat carboxylase oxygenase (RuBisCo, inside the chloroplast) and carbonic anhydrase (CA, right figure only) catalyze  $CO_2$  and COS fixation.

COS isotope discrimination during plant uptake could provide useful information on the uptake process and its response to environmental factors. The discrimination against CO34S (‰) is defined in Eq. (3), where 32k and 34k are the reaction rate coefficients for uptake of CO32S and CO34S, respectively:

$$^{34}\Delta = 1 - \frac{^{34}k}{^{32}k}. (3)$$

Isotope discrimination occurs both during diffusion of COS into the leaf and due to the preferential hydrolysis of lighter isotopologues by CA. Similar to the model developed by Farquhar et al. (1982) for  $^{13}$ CO2 discrimination during photosynthesis, the net CO34S discrimination during plant uptake ( $^{34}\Delta$ ) can be expressed as a function of the ratio of COS mole fraction at the site of assimilation (the end-point), in the mesophyll cell ( $C_m^S$ ) versus the COS mole fraction in ambient air ( $C_a^S$ ) (Davidson et al., 2022):

$$^{34}\Delta = \bar{a} + (h - \bar{a})\frac{c_m^S}{c_s^S},\tag{4}$$

where  $\bar{a}$  is the fractionation occurring during diffusion of COS into the leaf up to the mesophyll cell, which incorporates leaf boundary layer (BL) diffusion, stomatal diffusion and gas-liquid interface dissolution and diffusion, and h is the S isotope fractionation during fixation by the enzyme carbonic anhydrase (CA).

 $C_m^S$  has been suggested to be close to zero in C3 plants (Stimler et al., 2011; Stimler et al., 2012). When  $C_m^S$  = 0, Eq. (4) reduces to  ${}^{34}\Delta = \bar{a}$ , thus  ${}^{34}\Delta$  is caused solely by diffusion differences between CO32S and CO34S ( $\bar{a}$ ) through the stomata and up to the mesophyll. Binary molecular diffusion of COS in air is theoretically expected to provide a  ${}^{34}\Delta$  value of around 5 ‰, because of the differences in molecular masses between the different COS isotopologues (Angert et al. 2019). However, this may be a too crude simplification of the diffusion processes taking place. When including stomatal diffusion, leaf BL diffusion, and gas—liquid phase diffusion in the mesophyll cell, Davidson et al. (2022) calculated an overall diffusion fractionation value of  $\bar{a} = 1.6 \pm 0.1\%$  for  ${}^{34}$ S.

Still, it is not known whether the COS mole fraction in the mesophyll always reaches values close to zero, especially for C4 species, in which CA activity is lower (Stimler et al., 2011). In this case, values for the enzymatic

https://doi.org/10.5194/egusphere-2025-215 Preprint. Discussion started: 10 February 2025 © Author(s) 2025. CC BY 4.0 License.

100

105

110

115

120

125

130

fractionation during COS fixation by CA (h) are needed to calculate  $^{34}\Delta$ . Davidson et al. (2022) determined an enzymatic fractionation for  $^{34}$ S, h, of  $15 \pm 2$  ‰ from experiments in which the plants were exposed to high CO2 and COS mole fractions.

The observed  $^{34}\Delta$  values, measured in C3 and C4 species by Davidson et al. (2022), during their series of closed-chamber experiments, were  $1.6 \pm 0.1$  ‰ and  $5.4 \pm 0.5$  ‰, respectively, at ambient COS and CO2 mole fractions. Here, the higher discrimination value for C4 species likely reflects the lower CA activity, leading to higher  $c_m$  and therefore an influence of b on the observed discrimination.

To date, Davidson et al., (2021) and Davidson et al., (2022) are the only studies that have determined COS isotope discrimination during plant uptake, and they used a closed-chamber approach. As mole fractions of CO2 and COS change during experiments with closed chambers, there is a potential risk that feedback processes on stomatal conductance and other metabolic processes may contribute to the observed discrimination and hence the results may not reflect typical leaf conditions. With flow-through chambers, conditions can be monitored online and kept stable throughout the entire experiment, also allowing for easier repetition of the experiments.

In this work we used flow-through plant chambers, closely monitored to maintain stable conditions, to perform joint measurements of COS and CO2 fluxes in C3 and C4 species and at a range of PAR. We determined the isotope discrimination of COS uptake against CO34S and CO2 uptake against 13CO2 and C12O18O (34Δ, 13Δ, and 18Δ). The joint COS and CO2 measurements allowed investigating the relationship between COS and CO2 isotope effects, where the CO2 data provide additional information for validating the experimental setup and the plant behavior.

**2. Methods**

**2.1. Plant materials and growing conditions**

Experiments were conducted with the C3 plant sunflower (*Helianthus Annuus* "Sunsation") and the C4 plant papyrus (*Cyperus papyrus*). Sunflower plants in the flowering stage were obtained at a local garden center. In the case of papyrus, three large stems with leaves were carefully cut using a sharp razor, from a larger shrub growing in the tropical greenhouse at Wageningen University and Research (WUR). These leaves were transported with their cut stem in water to the lab and kept in water throughout the chamber measurements. The sunflower plant and papyrus cuttings were kept under a lamp with a solar-like spectrum (*ca.* 400 µmol m-2 s-1 PAR, LED growth light SMD2835, Ortho, China) before experiments started and watered sufficiently before and during the measurements. Leaf surface area of sunflower and papyrus were measured after the experiments using a LI-3100 (Li-Cor, Lincoln, NE, USA). This instrument was calibrated using a metal disk with a surface area of exactly 50.00 cm2.

**2.2. Whole plant gas exchange system**

Gas exchange experiments were conducted at Wageningen University and Research (WUR) using a custom-built whole plant chamber that was developed for estimating net photosynthetic CO2 assimilation and transpiration (Lazzarin et al., 2024). The main component is a flow-through plant chamber, which can be fed with different gas mixtures. Two analyzers were used to measure in- and outgoing mole fractions and we used an add-on module for discrete air samples (Fig 2.).

145

Figure 2. Schematic overview of the setup to determine CO2 and COS photosynthetic isotope discrimination by coupling a custom-built plant chamber to a LI-7000, a QCLS and a system to fill up gas canisters for posterior isotope analysis with IRMS. MFC: mass flow controller; QCLS: Quantum Cascade Laser Spectrometer. CO2 and COS were mixed into humidified synthetic air and introduced into the plant chamber. The in- and outflowing airstreams of the chamber (airin and airout) were measured by both the LI-7000 and QCLS instruments. Air was dried using Mg(ClO4)2 before the QCLS and when taking a sample for isotope analysis.

The plant chamber was made of clear plexiglass lined with a FEP foil (Holscot Europe, Breda NL) to prevent water from sticking to the chamber walls. The chamber had a diameter of 29 cm, and the height was either 18 or 27 cm, depending on the plant size. To ensure proper air mixing and leaf boundary layer reduction, three SanAce40W ventilators (type 9WL0424P3J001, Sanyo120 Denki, Philippines) were placed in a circular pattern at the bottom of the chamber. Fan speed was controlled with a SanAce PWM controller. The entire chamber was placed inside a 63x63 cm2 enclosure with white reflective walls that ensured uniform horizontal light distribution. Air temperature inside the plant chamber was measured with a LM35 temperature sensor (Texas Instruments). Temperature of the plant chamber was controlled using heating cables positioned around the outside of the plant chamber (in combination with a PID controller) and two 12V computer fans were used to provide airflow and cooling around the plant chamber. Light was

155

160

165

170

175

180

provided by LED lighting mounted above the chamber with a spectrum resembling sunlight (artificial sunlight research modules generation 2, Specialty Lighting Holland B. V., Breda, the Netherlands). PAR was quantified during the experiments just above the chamber using a handheld PAR sensor (LI-190, Li-Cor, Lincoln, NE, USA). Plants were placed in the chamber, and the bottom two plexiglass panels were closed around the stem of the plant and sealed it with Terostat RB VII, ensuring that the plant was isolated from the soil or water (in the case of the papyrus), and making sure the chamber was leak-free. Two pictures of the plant chamber are shown in Appendix A, Fig. A2.

Synthetic air humidified with a temperature-controlled water bubbler (dew point temperature 17 °C) was mixed with pure CO2 using mass flow controllers (MFC), to reach the desired CO2 and H2O mole fractions. Subsequently, COS from a cylinder with 700 ppb COS in synthetic "zero" air was supplied to the mix using a MFC to establish the target COS mole fractions of approximately 2 ppb. The flow rate of the total (combined) air mixture into the chamber was controlled by a MFC to around 8 L min-1, depending on the experiment conducted. The COS and CO2 isotopic composition of the ingoing air was determined using the methods described in 2.5 and the values are provided in Table 1.

Table 1. Isotope composition of the inlet gas ( $air_{in}$ ) supplying the plant chamber determined from samples collected in canisters and analyzed with IRMS.

| Plant     | δ 34 S COS (‰) | δ 13 C CO 2 (‰) | δ 18 O CO 2 (‰) |
|-----------|---------------------------|---------------------------------------|---------------------------------------|
| Sunflower | $11.9 \pm 1.2$            | $-23.1 \pm 0.1$                       | $15.5 \pm 0.1$                        |
| Papyrus   | $12.1 \pm 0.5$            | $-23.0 \pm 0.1$                       | $15.9 \pm 0.1$                        |

The CO2 and H2O mole fractions of both the in-going air (airin, reference line) and the outgoing air (airout, sample line) of the chamber were analyzed with a LI-7000 infrared gas analyzer (LI-COR Biosciences, Lincoln, Nebraska, USA). CO2 and COS mole fractions of the airin (reference) and airout (sample) lines were also measured with a QCLS from the Center for Isotope Research, Rijksuniversiteit Groningen (CIO-RUG). The QCLS used a 50 mL min-1 flow and was manually switched between airin, airout and calibration cylinders. The air entering the QCLS was dried with magnesium perchlorate (Mg(ClO4)2) dryers. Calibration of the QCLS was performed at least twice a day using the working standards from the CIO-RUG, which are calibrated against NOAA-certified cylinders. Possible instrumental baseline drift during the experiments was corrected by measuring pure nitrogen (N2) multiple times during the experiment. For a detailed description of the QCLS instrument and calibration procedures, see Kooijmans et al. (2017).

Samples for isotope analysis were taken in 6 L evacuated Silonite canisters (ENTECH, type: PN: 29- 10622) that were filled to ambient pressure. Sampling was done through a Mg(ClO4)2 dryer and a filter, and the flow into the canisters was regulated using a manual flow controller. The dryer was changed after every two samples. Sampling for COS and CO2 isotope composition started after ingoing and outgoing concentrations had stabilized, to ensure stable rates of photosynthesis, respiration, and COS assimilation. The stability of these fluxes, prior to sampling, was assured by checking the online data of the QCLS and LI-7000.

**2.3. Experimental conditions**

For all experiments the chamber was supplied with air mixtures with [COS] = 2300-2400 ppt, and  $[CO_2] = 430-440$  ppm at a flow rate of 8.1 L min-1, giving an air residence time of around 1.5–2 min. Temperature in the chamber was

190

200

205

24.6-25.0 °C in sunflower experiments and 25.7-25.9 °C in papyrus experiments. Light intensity was sequentially set to PAR = 400, 600, 200, and 0  $\mu$ mol m-2 s-1, allowing time after each light setting for plant adjustment, uptake flux stabilization and subsequent isotope sampling. Measurements at PAR 600  $\mu$ mol m-2 s-1 were not performed with the papyrus due to time constrains. At the start of each experiment with a new plant, two samples were taken of the ingoing air (airin). Samples were collected in 6 L canisters from airin (at the start of each experiment with a new plant) and airout (at each light setting). For the dark measurements chamber light was switched off and the chamber was covered with a blanket.

**2.4. Uptake flux calculations**

Both CO2 and COS net uptake fluxes ( $A^s$  in pmol m-2s-1 and  $A^c$  in µmol m-2s-1) were calculated using Eq. (5) (which shows the calculation for COS):

$$A^{s} = \frac{u_{e}}{S} \left( c_{e}^{s} - c_{a}^{s} \frac{1 - w_{a}}{1 - w_{a}} \right), \tag{5}$$

where  $u_e$  is the molar flow of air entering the chamber (mol air  $s^{-1}$ ), S is the leaf area (m2), and  $w_e$  and  $w_a$  (mol of H2O mol air-1) are the mole fractions of water vapor in airin and airout,  $c_e^s$  and  $c_o^s$  (pmol COS mol air-1) are the [COS] in airin and airout, respectively.

[revised manuscript text omitted]

$$\Delta = \frac{\xi(\delta_{out} - \delta_{in})}{1000 + \delta_{out} - \xi(\delta_{out} - \delta_{in})},\tag{9}$$

where  $\delta_{in}$  and  $\delta_{out}$  are the isotope compositions of the gas entering and leaving the chamber, respectively, for the gas of interest ( $\delta^{13}$ C,  $\delta^{18}$ O in CO2, or  $\delta^{34}$ S in COS).  $\xi$  is calculated as:

$$\xi = \frac{c_{in}}{c_{in} - c_{out}},\tag{10}$$

where  $c_{in}$  and  $c_{out}$  are the mole fractions of the gas of interest (in our case CO2 or COS), entering and leaving the chamber, respectively. At the start of each experiment, two canister samples were collected from the chamber inlet and their average was used to characterize  $air_{in}$  ( $c_{in}$  and  $\delta_{in}$ , Table 1), which was assumed constant over the experiment as it was supplied from a cylinder.

260

265

275

The errors on the measured mole fractions and isotope ratios were propagated to the isotope discrimination values ( $\Delta$ ); details are provided in the supplementary material.

Table 2. Isotope composition of the inlet gas (airin) supplying the plant chamber determined from samples collected in canisters and analyzed with IRMS.

| Plant     | δ 34 S COS (‰) | δ 13 C CO 2 (‰) | δ 18 O CO 2 (‰) |
|-----------|---------------------------|---------------------------------------|---------------------------------------|
| Sunflower | $11.9 \pm 1.2$            | $-23.1 \pm 0.1$                       | $15.5 \pm 0.1$                        |
| Papyrus   | $12.1 \pm 0.5$            | $-23.0 \pm 0.1$                       | $15.9 \pm 0.1$                        |

**3. Results and Discussion**

**3.1. COS and CO2 uptake fluxes**

In experiments with both plant species there was a net uptake of COS under all light conditions, including dark (Fig. 3b). Mean COS uptake fluxes in the light were  $74.2 \pm 1.5$  pmol m-2 s-1 and  $109.2 \pm 5.5$  pmol m-2 s-1 for sunflower and papyrus, respectively, and uptake fluxes did not vary strongly for different light conditions. Note that samples in the dark were taken sequentially, when plant conditions were still adjusting. Therefore, these samples were not treated as duplicates. As hydrolysis of COS, catalyzed by CA, is a light-independent reaction, COS assimilation can continue as long as the stomata are open (Protoschill-Krebs et al., 1996). Previously reported COS uptake fluxes at canopy- or ecosystem scale usually range between 30 and 60 pmol m-2 s-1 (Cho et al., 2023; Kooijmans et al., 2017; Commane et al., 2015; Billesbach et al., 2014), with some higher reported uptake fluxes around 80 to 100 pmol m-2 s-1 (Asaf et al., 2013; Spielmann et al., 2023). Thus, our measured COS uptake fluxes are at the high end of the spectrum, which may be due to the high ambient COS mole fraction inside the chamber.

Figure 3. AC (CO2 uptake flux, panel a, in µmol m-2 s-1) and AS (COS uptake flux, panel b, in pmol m-2 s-1) versus (PAR, µmol m-2 s-1) for sunflower (orange stars) and papyrus (green circles). Flux values for PAR > 0 are means ± 1 standard error (SE) (n = 2), where 1 SE was obtained using error propagation (see supplementary materials), flux values for PAR = 0 reflect individual measurements. Errors are only displayed when larger than the symbols.

For CO2, both sunflower and papyrus performed CO2 respiration in the dark and photosynthesis in the light, at a net rate that increased with PAR (Fig. 3a). Mean CO2 uptake fluxes in light conditions were  $6.7 \pm 1.7 \mu mol m^{-2}$

285

290

295

300

305

310

 $s^{-1}$  for sunflower and  $11.7 \pm 2.2 \,\mu\text{mol} \, \text{m}^{-2} \, \text{s}^{-1}$  for papyrus (Fig. 3a). These photosynthesis rates match that of sunflowers of Tezera et al. (2008) under their low-light condition experiments (in the least drought-exposed conditions).

At all light intensities (PAR>0), CO2 uptake rates were larger in papyrus than in sunflower, matching expectations for C4 vs. C3 photosynthesis (Farquhar & Lloyd, 1993). The photosynthesis rates for papyrus are comparable with previous measurements, conducted under low-light conditions. Our measurements can be classified as relatively low-light, because although the PAR measured at the top of the chamber reached 400 μmol m-2 s-1 at the highest setting for the C4 experiments, the PAR that was received by the plant leaves was likely lower, especially considering that some leaves were (partially) shaded or received diffused light, reflected off the outer enclosure walls. Ubierna et al., (2013) also found CO2 assimilation rates of around 10 μmol m-2 s-1 for PAR levels of 500 μmol m-2 s-1 in three C4 species, *Zea mays, Miscanthus* x *giganteus* and *Flaveria bidentis*, under varying light conditions between 0 and 2000 μmol m-2 s-1. Their results are similar to our measured CO2 uptake fluxes of between 9.4 μmol m-2 s-1 (200 PAR) and 14.0 μmol m-2 s-1 (400 PAR).

At PAR = 600  $\mu$ mol m-2 s-1, LRU (Eq. (1)) was 2.3 for sunflower and at PAR = 400  $\mu$ mol m-2 s-1, LRU values were 3.0 and 1.6 for sunflower and papyrus, respectively (see Table 2.). As PAR decreased to 200 µmol m-2 s-1, LRU increased to 5.2 for sunflower and 3.0 for papyrus. The increase in LRU at low light was due to a decrease in CO2 uptake fluxes while the COS uptake remained roughly constant. In the dark, LRU values were negative, up to −16.0 for sunflower, as COS uptake by the plant continued while CO2 was being respired. Our LRU values are higher than those found by Stimler et al. (2011) and higher than the usually reported median LRU value of 1.7 (Whelan et al., 2018), which may be due to our relatively low-light experiments. Yet, previously reported LRU values vary between 0.7 and 6.2, and Stimler et al. (2011) also reported a higher LRU for C4 compared to C3. Our slightly high LRU values could also be due to the higher than ambient COS mole fractions (of around 2ppb) that the plants were exposed to during our experiments. Davidson et al. (2022) reported LRU values or 0.7 and 1.7 for C3 and C4, respectively for experiment with ambient COS mole fractions, and LRU values of 2.4 and 1.0 for C3 and C4 for plants exposed to 2900 ppm CO2 and 3.4 ppb COS. Thus, exposure to higher COS mole fractions could influence LRU, however, more research is needed to quantify this effect. Furthermore, recent research has shown that LRU can differ across species and vary with environmental conditions, especially light availability and VPD (Kooijmans et al., 2019; Spielmann et al., 2023). The exact mechanism for this varying LRU is still not completely understood (Whelan et al., 2018; Wohlfahrt et al., 2023).

Figure 4 shows the CO2 uptake flux ( $\mu$ mol m-2 s-1) versus  $C_i^C/C_a^C$  ratio, which increases with decreasing CO2 uptake flux for both species. The species differences in CO2 uptake flux are consistent with the results presented by Stimler et al. (2011). Our measured  $C_i^C/C_a^C$  for sunflower compares well with previous values for sunflower of 0.8 found by Tezara et al. (2008). The  $c_i/c_a$  for papyrus is generally high for a C4 species, for which values usually range around 0.4, but could again be explained by the low-light conditions, as previously observed by Ubierna et al., (2013). The higher than usual  $C_i^C/C_a^C$  could also be explained by the fact that we measured entire plants, of which some leaves were partly shaded.

Figure 4.  $C_t^C/C_a^C$  plotted against  $A^C$  (CO2 uptake flux in  $\mu$ mol  $m^{-2}$  s-1), for sunflower (stars) and papyrus (circles). Colors indicate PAR levels ( $\mu$ mol  $m^{-2}$  s-1)

**315 3.2 CO34S discrimination**

Table 2 shows the isotopic discrimination for COS ( $^{34}\Delta$ ) and CO2 ( $^{13}\Delta$ ,  $^{18}\Delta$ ), and accompanying data for the different light treatments. In contrast to the CO2 isotope discrimination (Sect. 3.3),  $^{34}\Delta$  did not show a trend with COS uptake flux and PAR (Fig. 5),  $C_i^S/C_a^S$  (Fig. 6) or species. The average  $^{34}\Delta$  values in light conditions (PAR>0) were  $3.4 \pm 0.8$ % for sunflower and  $2.6 \pm 0.3$ % for papyrus (see Table 2). For sunflower in dark conditions, we found a  $^{34}\Delta$  of 4.7% for the first sample and 1.3% for the second sample, giving an average  $^{34}\Delta$  of  $3.0 \pm 2.3$ %. The COS uptake flux for papyrus in dark conditions decreased drastically to the point that  $^{34}\Delta$  could no longer be estimated with confidence.

Table 3. Photosynthetic discrimination (mean  $\pm 1\sigma$ , n=2), COS and CO2 uptake fluxes ( $A^S$  and  $A^C$ ), LRU,  $C_i^S/C_a^S$  and  $C_m^S/C_a^S$  for sunflower and papyrus, for each PAR level. The uncertainties were calculated as the standard deviation of the mean and the student's t-distribution, with 60% confidence interval and 1 (=n-1) degree of freedom. Values without stated uncertainty are single sample measurements (in the case of isotope discrimination values) or have an uncertainty smaller than 0.01(in the case of  $A^S$ ,  $C_i^S/C_a^S$  and  $C_m^S/C_a^S$ ).  $A^S$  at PAR = 0 for papyrus was too small for calculating  $^{34}\Delta$ .

| Plant     | PAR
(μmol
m -2 s -1 ) | 3 4 ∆
(‰) | 13 A (%0) | 18 ∆ (‰) | A S
(μmol
m -2 s -1 ) | (pmol m -2 s -1 ) | LRU | $C_i^S/C_a^S$ | $C_m^S/C_a^S$ |
|-----------|---------------------------------------------------|-------------------------|------------------|---------------------|--------------------------------------------------------------|-----------------------------------------|-----|---------------|---------------|
| Sunflower | 200                                               | 3.6 ± 1.2               | 32.4
± 1.1    | 148.7 ± 0.7         | 72.1                                                         | 4.42                                    | 5.2 | 0.50          | 0.11          |
| Sunflower | 400                                               | 3.7                     | 24.9             | 83.6                | 72.3                                                         | 6.86                                    | 3.1 | 0.52          | 0.07          |
| Sunflower | 600                                               | 2.8 ± 0.6               | 23.6
± 1.2    | 63.8 ± 0.9          | 74.9                                                         | 8.81                                    | 2.3 | 0.62          | 0.04          |

| a Sunflower | 0   | 3.0 ±     | -             | -          | 59.0 ±      | -     | -   | 0.45        | -    |
|------------------------|-----|-----------|---------------|------------|-------------|-------|-----|-------------|------|
|                        |     | 2.3       |               |            | 1.31        |       |     |             |      |
| Papyrus                | 200 | 2.5       | 21.8          | 79.4       | 108.6       | 9.36  | 3.0 | 0.39        | 0.05 |
| Papyrus                | 400 | 2.6 ± 0.4 | 18.9
± 3.4 | 49.4 ± 0.4 | 105.9       | 14.01 | 1.7 | 0.58        | 0.03 |
| Papyrus                | 0   | -         | 1             | -          | 24.6 ± 13.1 | -     | 1   | 0.72 ± 0.16 |      |

<sup>aThere was no uptake of  $CO_2$  at PAR = 0

Figure 5. Plant COS isotope discrimination ( $^{34}\Delta$ ) plotted against  $A^S$  (COS uptake flux in pmol  $m^{-2}$  s $^{-1}$ ) for sunflower (stars) and papyrus (circles). Colors indicate PAR levels ( $\mu$ mol  $m^{-2}$  s $^{-1}$ ).

340

345

350

355

Figure 6. Plant COS isotope discrimination ( $^{34}\Delta$ ) against the ratio of internal versus  $C_i^S/C_a^S$ , for sunflower (stars) and papyrus (circles). Colors indicate PAR levels ( $\mu$ mol  $m^{-2}$  s $^{-1}$ ).

To further investigate this lack of variability in  $^{34}\Delta$ , we examine the variability in  $C_i^S/C_a^S$  and  $C_m^S/C_a^S$  as a function of PAR (Table 2). We observed a slight increase of  $C_i^S/C_a^S$  with PAR, indicating that the stomata were perhaps not at their maximum opening at PAR $\leq$ 400, which was also suggested by the CO2 assimilation and isotope discrimination results (Figs. 4 and 7). However,  $C_m^S/C_a^S$  was rather stable at low values around 0.1– 0.23 over the various PAR levels and did not differ substantially between sunflower and papyrus. This lack in variability in  $C_m^S/C_a^S$  could explain the absence in variability in  $^{34}\Delta$  across the different light settings and between the two measured species, as previous studies (Stimler et al., 2011; Davidson et al., 2021; Davidson et al., 2022) attribute the differences in isotope discrimination between C3 and C4 species to differences in  $C_m^S/C_a^S$ .

Angert et al. (2019) estimated a value for  $^{34}\Delta$  during COS plant uptake of around 5 ‰ (based on binary diffusion theory), and experiments presented by Davidson et al. (2021) and Davidson et al. (2022) yielded  $^{34}\Delta$  values of  $1.6 \pm 0.1$  ‰ for  $C_3$  and  $5.4 \pm 0.5$  ‰ for  $C_4$  species. These are the only studies on COS isotope discrimination during plant uptake that have been conducted to date. Our results differ from these measurements, and we did not find statistically different  $^{34}\Delta$  values between our  $C_3$  and  $C_4$  species. The  $^{34}\Delta$  of 2.8 to 3.7 ‰ that we measured for sunflower is in between the  $^{34}\Delta$  for  $C_3$  found by Davidson et al. (2021; 2022) and the theoretical estimate of Angert et al. (2019). However, all  $^{34}\Delta$  estimations are roughly in the same range, which is reassuring given that different measurement techniques were used (flow-through chamber compared to closed-chamber).

The benefit of using a flow-through system is that stable environmental conditions inside the chamber can be maintained during the experiment. In contrast, in a closed chamber, CO2 and COS mole fractions will decrease due to plant uptake, which can be problematic when the experiment runs over long periods of time. Furthermore,

365

370

375

380

transpiration by the plant will increase the water vapor mole fraction in the chamber, which might affect stomatal opening and therefore also the isotope fractionation.

**3.3 CO2 isotope discrimination**

**3.3.1 13CO2 discrimination**

In both sunflower and papyrus, 13\(\Delta\) increased as the CO2 uptake flux decreased, with decreasing PAR (Fig. 7). Average 13Δ in sunflower was between 23.6 and 32.4 ‰ (Table 2), which is within the range of values expected for C3 photosynthesis (Farquhar et al. 1982, Kohn 2010, Cernusak et al. 2013, Wingate et al., 2007). However, in papyrus, 13Δ was between 18.9 and 21.8 ‰; much larger than the expected 3–6 ‰ for C4 species operating at optimal conditions (Farquhar et al 1983; Cerling et al. 1997; Kubásek et al., 2013; Ellsworth and Cousins, 2016; Eggels et al., 2021). As previously explained, our measurements were performed at low light intensities (PAR≤400 µmol m-2 s-1), which resulted in moderately low photosynthetic rates (9.3–14.0 μmol m-2 s-1). In C4 species, 13Δ has been shown to increase at low light to values as large as 8-17‰, when PAR = 50-125 µmol m-2s-1 (Ubierna et al. 2013, Pengelly et al. 2010, Kromdijk et al. 2010) and photosynthetic rates were small ( $<5 \,\mu$ mol m-2s-1). Our  $^{13}\Delta$  values for papyrus are still larger than these previous reports at low irradiance, suggesting that processes other than photosynthesis might have affected the measurements. Upward transport of water dissolved CO2 in the transpiration stream has been shown in tree stems (Aubrey and Teskey, 2009; Bloemen et al. 2013) and in papyrus culms (Li and Jones, 1995). We measured detached papyrus leaves submerged in water. This setting could have facilitated the transport of water dissolved CO2 into the leaf chamber, particularly because papyrus leaves have numerous vascular bundles surrounded by large air cavities (Plowman, 1906). Water dissolved CO2 would presumably have near-ambient air  $\delta^{13}$ C values – enriched compared to tank CO2 supplied to the chamber air -, and therefore if released in the plant chamber would artefactually increase  $^{13}\Delta$ .

Figure 7. Variation of photosynthetic discrimination against  $^{13}CO_2$  ( $^{13}\Delta$ , panel a) and  $CO^{18}O$  ( $^{18}\Delta$ , panel b) as a function of  $A^C$  ( $CO_2$  uptake flux in  $\mu$ mol  $m^{-2}$  s-1) for sunflower (stars) and papyrus (circles). Colors indicate PAR levels ( $\mu$ mol  $m^{-2}$  s-1).

**3.3.2 C16O18O discrimination**

390

395

400

From Fig. 7, we observe a negative relationship between  $^{18}\Delta$  and CO2 uptake flux, similar to  $^{13}\Delta$ . The average  $^{18}\Delta$  values of sunflower range between 63.8 and 148.7 ‰ and the average  $^{18}\Delta$  values of papyrus are between 49.4 and 79.4 ‰ (Table 2). Thus, the  $^{18}\Delta$  of papyrus is clearly lower than that of sunflower.  $^{18}\Delta$  mostly reflects the exchange of  $^{18}O$  between CO2 and leaf water (Francey and Tans; Yakir, 1998; Adnew et al., 2020). The lower  $^{18}\Delta$  in C4 species likely indicates the incomplete equilibrium between CO2 and leaf water, because of the reduced CA activity in C4 species compared to most C3 species (Gillon and Yakir, 2000).

A negative correlation of  $^{18}\Delta$  with CO2 assimilation and light intensity, as well as lower  $^{18}\Delta$  in C4 species was also found by Stimler et al. (2011). For their C3 plants, they found an  $^{18}\Delta$  which ranged between around 40 and 240 ‰, where the highest values were found at the lowest CO2 uptake fluxes. For C4 species, Stimler et al. (2011) found an  $^{18}\Delta$  between 10 and 50 ‰. Seibt et al. (2006) also found large variations in  $^{18}\Delta$  during CO2 uptake by *Picea sitchensis*, and a correlation with PAR. They too measured the largest  $^{18}\Delta$  discrimination at dusk and dawn, when light intensity was lowest.

The relation between the COS uptake flux and  $^{18}\Delta$  can also be analyzed, since both depend on the same diffusion pathway and CA activity (Stimler et al., 2011). Stimler et al. (2011) observed a clear negative correlation between  $^{18}\Delta$  and COS uptake flux, with a larger change in  $^{18}\Delta$  for  $C_3$  species, compared to  $C_4$ . Figure 8 shows  $^{18}\Delta$  against the COS uptake flux for our data. We do not observe such a correlation between  $^{18}\Delta$  and the uptake COS flux. However, our range in COS uptake flux for each species is small, as we found that the COS uptake flux did not change significantly when adjusting the light intensity. In the same range of COS uptake flux data, Stimler et al. (2011) did not find a strong trend in  $^{18}\Delta$  either.

Figure 8. 18 $\Delta$  (‰) plotted against  $A^S$  (COS uptake flux in pmol  $m^{-2}$  s-1) for sunflower (C3) and papyrus (C4), where the different symbols and colors indicate the plant types and PAR (µmol  $m^{-2}$  s-1).

**4 Conclusion**

This study presented measurements of COS and CO2 plant uptake fluxes and isotope discrimination factors  $^{34}\Delta$  of COS,  $^{13}\Delta$  and  $^{18}\Delta$  of CO2 and for sunflower (C3) and papyrus (C4). The experiments were conducted using a flow-

415

420

425

430

through gas exchange system, which is a new and different method compared to previously reported measurements of COS isotope fractionation during plant uptake (Davidson et al., 2021; 2022). The gas exchange system including the QCLS and LI-7000 instruments ensured stable chamber conditions, which were easy to monitor throughout the experiments.

Our study is the first to combine measurements of both COS and CO2 plant isotope discrimination, where the CO2 values provided additional information on the plant's behavior and their reactions to changes in environmental conditions. CO2 assimilation increased with increasing PAR level, consistent with previous results under similar conditions. However, the moderate to low-light conditions were limiting CO2 assimilation rate. Corresponding CO2 isotope discrimination values,  $^{13}\Delta$  and  $^{18}\Delta$ , were therefore higher at maximum capacity for CO2 assimilation rate. CO2 isotope discrimination as well as  $C_i^C/C_a^C$  were lower in papyrus than in sunflower, as expected and  $C_i^C/C_a^C$  decreased with light intensity for both species. Therefore, we conclude that both species were behaving normal, albeit not in the most optimal conditions for maximum capacity for photosynthetic CO2 assimilation.

In contrast to photosynthesis, COS assimilation was light-independent, which is expected since the hydrolysis reaction catalyzed by CA does not require light. The observed COS uptake flux was lower during the dark experiments, but not zero, indicating some residual stomatal opening. Our measurements also showed a constant  ${}^{34}\Delta$  across different light settings, which can be explained by the rather constant  $C_i^S/C_a^S$  and  $C_m^S/C_a^S$  values. Surprisingly,  ${}^{34}\Delta$  also did not differ significantly between papyrus and sunflower, whereas previous measurements (Davidson et al., 2022) did show a higher  ${}^{34}S$  isotope discrimination for C4 species. However,  $C_i^S/C_a^S$  and  $C_m^S/C_a^S$  were also not different between our measured C3 and C4 species, hence similar isotope discrimination is expected. Nevertheless, our values for  ${}^{34}\Delta$  are close to the previously reported values by Davidson et al. (2022), despite using a different experimental set-up and a different way to calculate the isotopic discrimination (Evans et al., 1986).

**Appendices**

**Appendix A: Supplementary figures**

Figure A1. CO2 and COS fluxes in  $\mu$ mol  $m^{-2}$   $s^{-1}$  and pmol  $m^{-2}$   $s^{-1}$ , respectively, calculated from the discrete samples that were analyzed on the mass spectrometer, plotted against the fluxes that were calculated from the online QCLS measurements. Uncertainty bars are  $\pm 1\sigma$ , obtained using error propagation of the measurement errors on all the components used during the flux

calculations (see supplementary materials). The errors are only depicted when they are larger than the symbols. The stars symbols are the sunflower data, and the circles are the papyrus data. The different color shadings indicate the varying PAR levels in µmol  $m^{-2}$  s-1. The black dashed line shows the one-to-one line, for reference. The two samples that clearly fall off the line in the CO2 plot were excluded from both the CO2 and COS dataset, as these sample canisters had possibly leaked or were contaminated with air other than the plant chamber air.

445 Figure A2. Pictures of the plant chamber, with sunflower (left) and papyrus leaves (right) inside. The chamber consists of two cylinders, connected to each other and to the upper and lower panels with Terostat RB VII. The plant pot and soil are kept outside of the chamber and the chamber is sealed onto the stem with Terostat as well. The black wires are automated (computer controlled) heating wires, ensuring constant temperature around the chamber.

**Appendix B: Gas exchange calculations for CO2 and COS**

450 We detail gas exchange equations of von Caemmerer and Farquhar (1981) for CO2 and adapt this theory to derive gas exchange parameters for COS. For assimilation rates and mixing ratios we adopt a nomenclature where the superscript c refers to CO2 and s to COS. For conductances the subscript represents the molecule of interest (w – water,  $c - CO_2$ , s - COS) and the superscript the type of conductance (t - total, b - boundary layer, s - stomata).

 $CO_2$  and COS assimilation rates ( $A^c$ ,  $A^s$ ,  $\mu$ mol  $CO_2$  m-2 s-1):

455
$$A^{c} = \frac{u_{e}}{S} \left( c_{e}^{c} - c_{a}^{c} \frac{1 - w_{e}}{1 - w_{a}} \right),$$

$$A^{s} = \frac{u_{e}}{S} \left( c_{e}^{s} - c_{a}^{s} \frac{1 - w_{e}}{1 - w_{a}} \right),$$
(B2)

where  $u_e$  is the molar flow of air entering the chamber (mol air s-1), S is the leaf area (m2),  $c_e^c$  and  $c_a^c$  ( $\mu$ mol CO2 mol air 1) are the [CO2] in the air entering and leaving the chamber, respectively, and  $c_e^s$  and  $c_a^s$  (pmol COS mol air 1) are the [COS] in the air entering and leaving the chamber, respectively.

460 Transpiration rate (mol H2O m-2s-1)

$$E = \frac{u_e}{S} \frac{w_a - w_e}{1 - w_a},\tag{B3}$$

where  $w_e$ ,  $w_a$  (mol of H2O mol air-1) are the mole fractions of water vapor in the air *entering* the chamber and in the chamber *air* (which equals to the air *out* of the chamber).

465 Total conductance to water vapor  $(g_w^t, \text{ mol H}_2\text{O m}^2 \text{ s}^{-1})$ :

$$g_w^t = E \frac{1 - \frac{w_i + w_a}{2}}{w_i - w_a},\tag{B4}$$

where (mol of H2O mol air-1) is the mole fraction of water vapor *inside* the leaf, which assuming saturation with water vapour at the leaf temperature ( $T_U$  °C) can be calculated:

$$w_i = \frac{0.61635e^{\frac{17.502T_l}{240.97+T_l}}}{P_a},\tag{B5}$$

470 where  $P_a$  (kPa) is atmosphere pressure in the chamber.

Stomata conductance to water  $(g_s^w, \text{ mol H}_2\text{O m}^{-2} \text{ s}^{-1})$  is:

$$g_s^w = \frac{1}{\frac{1}{g_t^w} - \frac{1}{g_b^w}},\tag{B6}$$

where  $g_b^w$  is the boundary layer conductance to water, a characteristic of each plant chamber, but often very large in well stirred chambers (a requisite for gas exchange).

Total conductance to  $CO_2(g_t^c, \text{mol } CO_2 \text{ m}^{-2} \text{ s}^{-1})$  and  $COS(g_t^s, \text{mol } COS \text{ m}^{-2} \text{ s}^{-1})$ :

$$g_t^c = \frac{1}{\frac{1.6}{g_s^w} + \frac{1.37}{g_b^w}},\tag{B7}$$

$$g_t^s = \frac{1}{\frac{1.94}{a_t^w} + \frac{1.56}{a_t^w}},\tag{B8}$$

where the coefficient 1.6 and 1.37 (mol H2O mol CO2-1) are the ratio of diffusivities of CO2 to water vapor in air, and in the boundary layer, respectively. The coefficients 1.94 and 1.56 (mol H2O mol COS-1) are the ratio of diffusivities of COS to water vapor in air, and boundary layer, respectively (Fuller *et al.*, 1966; Farquhar & Lloyd, 1993).

Concentration inside the leaf of CO2 ( $c_i^c$ , µmol CO2 mol wet air-1) and COS ( $c_i^s$ , pmol COS mol wet air-1)

 $A^c$  and  $A^s$  are determined with gas exchange with Eqs. (B1) and (B2), and can also be related to the [CO2] and [COS] inside the leaf with the equations:

485  $A^{c} = g_{t}^{c}(c_{a}^{c} - c_{i}^{c}) - E\frac{c_{a}^{c} + c_{i}^{c}}{2},$  (B9)

$$A^{s} = g_{t}^{s}(c_{a}^{s} - c_{i}^{s}) - E\frac{c_{a}^{s} + c_{i}^{s}}{2},$$
(B10)

where  $E \frac{c_a^c + c_i^c}{2}$  and  $E \frac{c_a^s + c_i^s}{2}$  are ternary corrections that accounts for the influence of transpiration on the diffusion of CO2 and COS into the leaf. Solving  $c_i^c$  from Eqn 9 and  $c_i^s$  from Eq. (B10) results in:

$$c_i^c = \frac{\left(g_t^c - \frac{E}{2}\right)c_a^c - A^c}{g_t^c + \frac{E}{2}},\tag{B11}$$

$$c_i^s = \frac{\left(g_t^s - \frac{E}{2}\right)c_a^s - A^s}{g_t^s + \frac{E}{2}}.$$
 (B12)

COS concentration in the mesophyll at the sites of CA ( $c_m^s$ , pmol COS mol wet air-1):

By analogy with the model for photosynthetic discrimination against 13CO2 (Farquhar et al., 1982; Farquhar & Cernusak, 2012) discrimination against CO36S (‰) during plant uptake can be described:

$$\Delta^{34}S = \frac{1}{1-t} \frac{a_{c_i^s}}{c_a^s} \frac{c_a^s - c_i^s}{c_a^s} + \frac{1+t}{1-t} \left[ a_m \frac{c_i^s - c_m^s}{c_a^s} + h \frac{c_m^s}{c_a^s} \right], \tag{B13}$$

where  $\overline{a_{c_i^s}}$  (‰) is the weighted discrimination for diffusion across the leaf boundary layer and inside the mesophyll, calculated as:

$$\overline{a_{c_i^s}} = \frac{a_b(c_a^s - c_s^s) + a_s(c_s^s - c_i^s)}{c_a^s - c_i^s},$$
(B14)

with  $c_s^s$ , the [COS] (pmol COS mol wet air-1) at the leaf surface, is

$$c_s^s = c_a^s - A^s \frac{1.56}{g_b^w}.$$
 (B15)

The t is a ternary correction factor calculated as (Farquhar & Cernusak, 2012):

$$t = \alpha_{ac} \frac{E}{2g_t^s},\tag{B16}$$

where  $\alpha_{ac} = 1 + \frac{\overline{\alpha_{c_l}^s}}{1000}$

505 The  $a_b$  (= 3.5%),  $a_s$  (= 5.2%), and  $a_m$  (= 0.5%) are fractionations for COS diffusion across the boundary layer, across the stomata, and due to COS dissolution and diffusion in water through the mesophyll, respectively (Davidson et al., 2022). h (=15 ± 2‰) is the fractionation during COS hydrolysis by CA (Davidson et al., 2022).

The  $c_m^s$  can be solved from Eqn 13 as:

$$c_m^s = \frac{(1-t) \cdot \Delta^{34} S \cdot c_a^s - \overline{a_{c_i^s}}(c_a^s - c_i^s) - (1+t) \cdot a_m \cdot c_i^s}{(1+t)(h-a_m)}.$$
(B17)

510 Because  $t \cong 0$ , then Eq. (B17) can be simplified to

$$c_m^s \cong \frac{\Delta^{34} S \cdot c_a^s - \overline{a_{c_i^s}}(c_a^s - c_i^s) - a_m \cdot c_i^s}{h - a_m}.$$
(B18)

Substituting in Eq. (B18) the  $\overline{a_{cs}}$  for its expression given in Eq. (B14) and rearranging terms result in:

$$c_m^s \cong \frac{c_a^s(\Delta^{34}S - a_b) + c_s^s(a_b - a_s) + c_i^s(a_s - a_m)}{h - a_m}$$
(B19)

Substituting in Eq. (B19) the fractionation factors by their values results in:

515
$$c_m^s \cong \frac{(\Delta^{34}S - 3.5)c_a^s - 1.7c_s^s + 4.7c_i^s}{14.5}, \tag{B20}$$
 where  $\Delta^{34}S$  (‰) can be experimentally determined during measurements of gas exchange as (Evans *et al.*, 1986):

$$\Delta^{34}S = \frac{c_e^S}{c_e^S - \frac{c_o^S}{c_o^S}} \frac{\delta_o^{34} - \delta_e^{34}}{1 + \delta_o^{34} - \frac{c_o^S}{c_o^S - \frac{c_o^S}{c_o^S}} (\delta_o^{34} - \delta_e^{34})},$$
(B21)

[revised manuscript text omitted]

---

## Author Response (AR2)

Baartman et al. Isotope discrimination of carbonyl sulfide (34S) and carbon dioxide (13C, 18O) during plant uptake in flow-through chamber experiments

Revisions 31-07-2025

**Reviewer's suggestions:**

"The paper has seen significant improvements since the initial draft. The authors have indeed worked significantly to enhance the narrative and contextualize their findings within the framework of existing studies. The method is also clarified. I only have minor comments that can improve the manuscript, based on the lines of the track change.

Line 21: Add the word "Irreversibly" catalyzed

This has been added

Line 32: Add a S to conductanceS

This has been added

Line 55: Add "For both C3 and C4 plants, for CO2, a negative relationship..."

This has been added

Line 56: Rephrase the first part of the sentence "The CO2 uptake ... plant," to ".. The Co2 uptake and C13 and C18 discriminations of sunflower have expected values for a C3 plant"

We rephrased this beginning of the sentence to "The CO2 uptake and 13CO2 and C16O18O discriminations of sunflower have expected values for a C3 plant,..."

Reply to reviewer 2 in reference to line 43: Please report these lines in the introduction to better explain the motivations.

We added the following to the sentence from lines 53 – 54 (tracked changes version of manuscript): "solar-induced chlorophyll fluorescence (SIF), near infrared reflectance of vegetation (NIRv) and inverse atmospheric modeling studies (Kettle et al., 2002; Ma et al., 2021; Remaud et al., 2022)."

And in line 57: "or, in the case of modeling studies, prior information on location and magnitude of the fluxes."

These additions are in line with our response to the comment from reviewer #2 on the introduction.

Line 125: The addition of "and as the reaction with CA is supposed to be irreversible" is inconsistant with lines 161 which explain that, in C4 plants, the CA activity is low. Cho et al. (2024) showed that CA activity depends on temperature and reaches a maximum at a specific temperature (<a href="https://bg.copernicus.org/articles/21/3735/2024/bg-21-3735-2024.html">https://bg.copernicus.org/articles/21/3735/2024/bg-21-3735-2024.html</a>). How does the dependency affect h?

We thank the reviewer for this question as this is still one of the uncertainties/questions in the field of COS isotope discrimination in plants. Davidson et al. (2022) mention that they suspect that COS is able to diffuse out of the leaf again in C4 plants as C1 increases due to the lower CA activity. This

means that part of the COS is not hydrolyzed "fast enough" by CA, COS builds up to higher concentrations inside of the leaf (compared to C3 plants) and may partly diffuse out again. This COS that diffuses out would then have some fingerprint of the CA discrimination against the heavier isotopologue  $^{34}$ S as the lighter one  $^{32}$ S would be preferred to go into the reaction. Hence the COS inside of the leaf would be slightly enriched in the heavier  $CO^{34}$ S, thus a stronger discrimination would be observed in a  $C_4$  plant than compared to a  $C_3$ . But this is still theoretical and needs to be confirmed with experimental data.

In equation (4), h represents the discrimination of CA against CO34S, which we assume to be a constant value, for now. But how much this term h influences the overall observed discrimination  $^{34}\Delta$  depends on the Cm/Ca ratio, which we think would be higher in C4 species, following the line of reasoning above.

In terms of temperature dependency, we expect that a higher temperature leading to higher CA activity would then lower the Cm/Ca ratio and following Eq. (4) would then also decrease the observed discrimination. And at a maximum CA activity, Cm/Ca would be close to zero, which leads to  $^{34}\Delta=\bar{a}$  (line 119 in current version of manuscript). Since we did not do a temperature response experiment ourselves, we choose not to go into the details of the CA activity – temperature dependency.

**Implemented changes:**

- → We removed the part "and as the reaction with CA is supposed to be irreversible" in order to avoid confusion.
- $\rightarrow$  We rephrased the sentence in lines 127-128, which now reads "In the case of non-zero  $C_m^S$ , enzymatic fractionation during COS fixation by CA (h) will affect the observed  $^{34}\Delta$  (Eq. (4))."

Part 2.4 As you show the values gsw in Table 2, the formula of gsw should also be shown in this part.

We agree that we need to provide calculations for gsw, but since we would then have to include three equations for full clearity and we do not wish for this paragraph to become too lenghty, we refer the reader now to Appendix B, Equations (B3), (B4) and (B5), where we provide all detailed calculations. We hope the reviewer agrees with this solution.

Line 449: For which kind of plants the mentioned values from Stimlers apply? C3 or C4?

We mentioned these data in a more specific way by adding "ranging between around 15 to 30 pmol m-2 s-1 for the  $C_4$  species maize, sorghum and amaranthus, under a light intensity of 500  $\mu$ mol m-2 s-1,..." (lines 328-329 in new version of the manuscript)

Appendix C: Why are the values of PAR from Davidson (2022) so low in the Table?"

We were also surprised by these very low light intensity in their experiments, but these are the numbers that Davidson et al. (2022) provide in their supplementary material. Since they only used one (small) lamp at the top of their chamber, and used species with a large leaf area, we expect that this is why the light intensity dimished so drastically within their chamber.

**Editor's suggestions:**

Line 111: Change "ribulose-1,5-biphosphate" to "ribulose-1,5-bisphosphate"

**This has been changed**

Line 394: Please also mention in the header of Table 2 that  $18\Delta$  indicates the apparent discrimination.

**This has been added**

Line 557: I think you meant "slightly higher LRU values"

**Indeed, this has been changed**

Line 565: Add "plants" after "C3 and C4"

**The word "plants" has been added**

**Additional edits:**

- Missing period was added in line 88.
- As per request from the editorial support, we changed the direction of page 15 back to
  portrait mode and rotated the table, so that we can still include the entire table with all the
  necessary information in the main text of the manuscript. We hope the layout works like this
  and otherwise we will discuss other options.
- We edited the reference list to be in compliance with the Copernicus formatting.